# Code Repair with LLMs gives an Exploration-Exploitation Tradeoff

**Hao Tang**
Cornell University
haotang@cs.cornell.edu

**Keya Hu**
Shanghai Jiao Tong University
hu_keya@sjtu.edu.cn

**Jin Peng Zhou**
Cornell University
jpzhou@cs.cornell.edu

**Sicheng Zhong**
University of Toronto
sicheng.zhong@mail.utoronto.ca

**Wei-Long Zheng**
Shanghai Jiao Tong University
weilong@sjtu.edu.cn

**Xujie Si**
University of Toronto
CIFAR AI Chair, Mila
six@cs.toronto.edu

**Kevin Ellis**
Cornell University
kellis@cornell.edu

## Abstract

Iteratively improving and repairing source code with large language models (LLMs), known as *refinement*, has emerged as a popular way of generating programs that would be too complex to construct in one shot. Given a bank of test cases, together with a candidate program, an LLM can improve that program by being prompted with failed test cases. But it remains an open question how to best iteratively refine code, with prior work employing simple greedy or breadth-first strategies. We show here that refinement exposes an explore-exploit tradeoff: exploit by refining the program that passes the most test cases, or explore by refining a lesser considered program. We frame this as an arm-acquiring bandit problem, which we solve with Thompson Sampling. The resulting LLM-based program synthesis algorithm is broadly applicable: Across loop invariant synthesis, visual reasoning puzzles, and competition programming problems, we find that our new method can solve more problems using fewer language model calls.

## 1   Introduction

An emerging paradigm for problem-solving with large language models (LLMs) is to have the language model *correct, repair, or debug* its initial outputs [1, 2, 3, 4, 5], which we refer to here as *refinement*. For example, when generating the source code of a program, refinement prompts the LLM with a buggy program it previously generated, potentially with diagnostic information such as a stack trace, then asks it to fix its code. This strategy alleviates the need for the LLM to predict perfect code on the first try, and approximates the iterative code-writing process of software development.

Complex programming problems often require several rounds of refinement, with each round requiring more LLM calls, each of which is stochastic, and so yields multiple possible outcomes. Therefore, this process generates a tree of possible programs (Fig. 1 left). This tree is infinitely deep, because every refinement can itself be further refined. It is also infinitely wide, because the LLM can return infinitely many possible refinements. Success should, in principle, depend on exactly what policy is used to expand this tree. Recent work [1, 6, 3] adopts simple expansion policies (e.g., breadth-first),

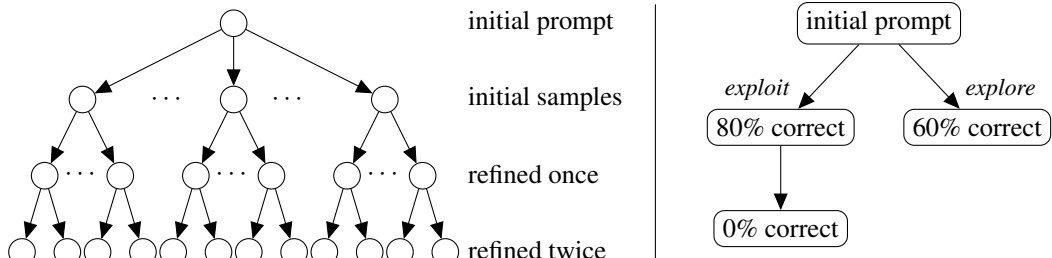

Figure 1: Left: The tree of possible refinements is infinitely deep and has infinite branching factor. Each node is a program and each edge is an LLM sample. Right: Explore-Exploit tradeoff for a search state after performing 3 node expansions. Exploit by sampling another child of a program that is nearly correct, or Explore by sampling a child of a program that has been expanded fewer times.

with mixed success: On close examination, the gains from refinement, at least with these basic policies, are marginal compared to repeatedly sampling i.i.d. programs from the initial prompt [3].

Our work here reanalyzes refinement in terms of a tradeoff between exploiting the refinement of programs that are closer to being correct (i.e., pass more testcases), and exploring programs that have been refined fewer times. Fig. 1 (right) diagrams this tradeoff for a tree after the first few node expansions. This exploration-exploitation tradeoff is made more challenging by the fact that every time we refine a program, we create another brand-new program, giving an ever-increasing set of possible actions (possible next refinements). Our problem is however not solvable by standard approaches to Monte Carlo Tree Search (MCTS)—a bandit-based node expansion policy—because our branching factor is infinite, our transition function is stochastic, and "rollouts" would demand a prohibitively expensive series of LLM calls to refine down to some maximum depth.

Our primary contribution is an algorithm for efficiently performing refinement, which we call *REx* (REfine, Explore, Exploit). *REx* should be seen as a strategy for conducting LLM-guided search that constructs and navigates a tree of refinements. We derive *REx* via a multiarmed bandit framing: different actions correspond to refining different programs; reward corresponds to the quality of a newly generated program; and maximizing discounted future reward corresponds to solving the programming problem in the minimum number of LLM calls.

The resulting algorithm is broadly applicable to LLM-based code generation tasks. We describe applications to competition programming problems, challenging software verification problems involving generating nonlinear loop invariants, and visual reasoning puzzles from the Abstraction and Reasoning corpus (ARC: [7]). Across every domain, *REx* solves more problems in fewer LLM calls, typically reducing the amount of API calls by a factor of 1.5x-5x. *REx* is also able to consistently solve a modest number of difficult problems that prove out-of-reach for other approaches.

## 2 Background: Bandits and Thompson Sampling

**Bandit problems** concern maximizing total reward given a set of actions, $\mathcal{A}$, each with an unknown reward distribution, $P(r|a)$, for each action $a \in \mathcal{A}$. At each time step $t$, an action $a_t$ is chosen and a reward $r_t$ is received. Bandit problems are challenging because they involve balancing exploration and exploitation: On the one hand, each action must be tried enough times to estimate its average reward (exploration), but asymptotically, only the actions with the highest expected reward should be chosen (exploitation). Actions are called *arms*, and taking an action is called *pulling* that arm.

**Thompson Sampling** is a framework for designing bandit strategies that performs probability matching: pulling an arm with probability equal to the odds that that arm is optimal (has the highest expected reward). It maintains probabilistic beliefs about the reward distribution of each arm, and updates these beliefs following Bayes rule. Writing $\theta_a$ for the parameters of the reward distribution for arm $a$, this means that $P(r|a) = P(r|\theta_a)$ (by definition), and that $\theta_a$ is treated as a random variable with a prior $P(\theta_a)$. Beliefs are updated after each reward, and the next arm can be selected by sampling $\theta_a$ from the posteriors, and pulling the arm with the highest (expected) reward. Concretely,

Thompson Sampling does the following at timestep $t$:

$$\text{Sample } \theta_a \sim P(\theta_a| \{r_t \; : \; a_t = a\}), \text{ for each } a \in \mathcal{A}$$

$$\left(\text{Do the above using Bayes Rule: } P(\theta_a| \{r_t \; : \; a_t = a\}) \propto P(\theta_a) \prod_{t:a_t=a} P(r_t|\theta_a)\right)$$

$$\text{Pull arm } a_t = \underset{a \in \mathcal{A}}{\arg \max} \, \mathbb{E}\left[r \mid \theta_a\right]$$

$$\text{Get reward } r_t \tag{1}$$

By choosing different priors and likelihoods, a variety of Thompson sampling strategies can be designed. For example, prior knowledge of the value of each arm can be embedded in $P(\theta_a)$, and constraints on the range of possible rewards can be embedded in $P(r|\theta_a)$.

## 3 Problem Statement and Assumptions

**Definitions: Specification, ($\vdash$).** We are given a programming problem to solve which comes equipped with an efficiently-checkable **specification $\Phi$**. Our objective is to construct a program, written $\rho$, which **satisfies the specification, written $\rho \vdash \Phi$**. Generally, the specification decomposes into a conjunction of more basic constraints, notated $\{\phi_k\}_{k=1}^K$, where

$$\Phi = \phi_1 \wedge \phi_2 \wedge \phi_3 \wedge \cdots \wedge \phi_K \tag{2}$$

For example, a common form for $\Phi$ is a collection of input-output examples, with each $\phi_k$ corresponding to a specific input-output pair. As another example, when synthesizing loop invariants for software verification problems, each $\phi_i$ corresponds to a different verification condition.

**Definition: Counterexamples.** Given a program $\rho$ and a specification $\Phi$ of the form $\phi_1 \wedge \phi_2 \wedge \cdots \phi_K$, a **counterexample** is a conjunct $\phi_k$ where $\rho \nvdash \phi_k$. We define the set of all counterexamples as

$$\text{COUNTEREXAMPLES}\left(\rho, \{\phi_k\}_{k=1}^K\right) = \{\phi_k \; : \; \rho \nvdash \phi_k\} \tag{3}$$

**Refinement.** Refining a program prompts an LLM to improve it in ways that are likely to cause it to satisfy the specification. We write $P_{\text{refine}}(\cdot|\rho, \Phi)$ to mean the distribution of possible refinements. Although the exact design of this distribution depends on the program synthesis domain, the refinement operator is generally implemented by prompting an LLM with a random counterexample:

$$P_{\text{refine}}(\rho' \mid \rho, \Phi) = \underset{\phi \sim \mathcal{U}(\text{COUNTEREXAMPLES}(\rho,\Phi))}{\mathbb{E}} \left[P_{\text{LLM}}\left(\rho' \mid \text{prompt}(\rho, \phi, \Phi)\right)\right] \tag{4}$$

**Heuristic measures of progress.** Our method will organize its search strategy so as to prioritize programs that appear to be on the right track toward satisfying the specification. In general we assume access to a blackbox heuristic estimator of program goodness, $h(\rho)$, with values ranging from zero to one. Within this paper we work with a very basic heuristic that simply reports the fraction of specifications that are satisfied:

$$h(\rho) = \frac{\sum_{k=1}^K \mathbb{1}\left[\rho \vdash \phi_k\right]}{K} \tag{5}$$

More complex heuristics are possible, such as those which weigh certain parts of the specification more heavily, or which apply more fine-grained notions of approximate correctness, such as having low edit distance to a target output.

## 4 *REx*: Refine, Explore, Exploit

At a high level, we frame this problem as an **arm-acquiring bandit**, meaning a bandit problem where new arms arrive over time [8]. Here each arm is a program, and "pulling" an arm corresponds to refining a program. Because refinement is stochastic, we receive a random reward after each refinement, depending on whether the newly generated program satisfies the specification. New programs arrive over time, because with each refinement, we generate a new program, hence the

framing as "arm-acquiring." Our approach derives from Thompson Sampling, a stochastic algorithm that pulls an arm with probability equal to the probability that it is the best arm. Therefore we need to calculate probabilistic beliefs of the optimality of a given arm, which will be biased by the heuristic function $h$ to prioritize programs with high heuristic value. The derivation of our Thompson Sampler will also automatically penalize programs that have been refined many times.

Concretely, we receive a reward of 1 if pulling an arm (refining a program) yields a new program that satisfies $\Phi$, and otherwise receive reward zero. Therefore the reward $r$ follows a Bernoulli distribution, whose parameter we will write $\theta_\rho$:

$$\theta_\rho = P(r = 1|\rho, \Phi) = \mathop{\mathbb{E}}_{\rho' \sim P_{\text{refine}}(\cdot|\rho, \Phi)} [\mathbb{1}\left[\rho' \vdash \Phi\right]] \tag{6}$$

As we solve this bandit problem using Thompson Sampling [9, 10], we need to maintain probabilistic beliefs about each arm's expected reward, $\theta_\rho$, which get updated with each refinement. This means we need a prior over $\theta_\rho$, for which we choose the Beta distribution, which is conjugate to the Bernoulli, yielding simple Bayesian updates [11]. We choose a prior that places more mass on $\theta_\rho$ whenever $h(\rho)$ is also large, effectively injecting heuristic knowledge into the prior:

$$P(\theta_\rho) = \text{Beta}(\alpha_\rho^{\text{prior}}, \beta_\rho^{\text{prior}}) \tag{7}$$

$$\alpha_\rho^{\text{prior}} = 1 + C \times h(\rho) \tag{8}$$

$$\beta_\rho^{\text{prior}} = 1 + C \times (1 - h(\rho)) \tag{9}$$

where $C$ is a hyperparameter. Large $C$ gives greedier behavior by increasing the importance of $h$.

The posterior beliefs over $\rho$ get updated whenever $\rho$ is refined. Assume the program (arm) has been refined (pulled) $N_\rho$ times, all with no success (no reward). Then the posterior is

$$P(\theta_\rho|N_\rho) \propto P(N_\rho|\theta_\rho)P(\theta_\rho) = (1 - \theta_\rho)^{N_\rho} \times \text{Beta}(\alpha_\rho^{\text{prior}}, \beta_\rho^{\text{prior}})$$

$$\propto \text{Beta}(\alpha_\rho^{\text{prior}}, \beta_\rho^{\text{prior}} + N_\rho)$$

$$= \text{Beta}(1 + C \times h(\rho), 1 + C \times (1 - h(\rho)) + N_\rho) \tag{10}$$

The above equation essentially defines *REx*. For each program we track its heuristic value and how many times it has been refined, and to select the next program to refine, we sample from the Beta distributions in Eq. 10 and refine the program $\rho$ with highest sampled $\theta_\rho$. An implementation is about ten lines of Python (see below and Appendix Alg 1). Refining an empty program means generating a new program from scratch. The heuristic value of an empty program is unknown so we set $C \times h(\texttt{null}) = 1 - C \times h(\texttt{null}) = \text{NULL} := 0$.

```python
def REx(problem, C):
    programs = {problem.empty_solution()}
    failed_cnt = defaultdict(lambda: 0)              # N_ρ in paper
    while True:
        program = max(programs, key=lambda p: np.beta(
            1 + C*p.heuristic_value,                  # 1 + C × h(ρ) in paper
            1 + C*(1-p.heuristic_value)+failed_cnt[p] # 1 + C × (1 − h(ρ)) + N_ρ
        ))
        new_program = program.refine(problem)         # ρ' ∼ P_refine(·|ρ, Φ)
        if is_solved(problem, new_program):           # ρ' ⊢ Φ in paper
                return new_program
        failed_cnt[program] += 1
        programs.add(new_program)
```

**Understanding the behavior of *REx*.** Intuitively, we begin with optimistic beliefs about the utility of refining a program. The strength of that optimism is controlled by $h$, with higher heuristic-value programs having higher initial estimates of $\theta_\rho$. With each unsuccessful refinement, we update our beliefs to be less optimistic, with the expected reward decaying toward zero with increasing number of failures:

$$\mathbb{E}\left[\theta_\rho \mid N_\rho\right] = \frac{1 + C \times h(\rho)}{2 + 2C + N_\rho} \tag{11}$$

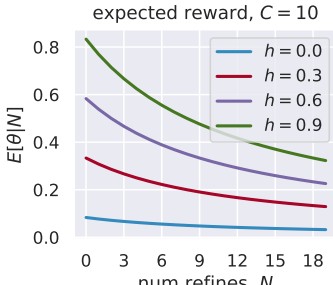 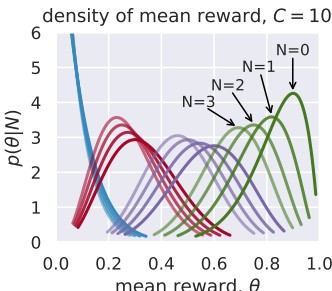 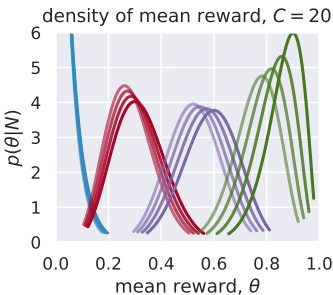

Figure 2: How the model's beliefs about the benefit of refining a program, $\theta$, change as we vary (1) $N$, the number of times it was previously refined, and (2) $h$, the heuristic estimate of how close we are to satisfying the specification (larger $h$ is better). Left: Expected benefit of refining decreases the more we refine, and asymptotically decays to zero (Eq. 11). Middle/Right: Posterior beliefs initially center around $h$ and shift toward zero with each additional refinement. Same colored curves show same values of $h$ for different values of $N$. The hyperparameter $C$ modulates the rate of decay with each additional refinement, and also affects the initial concentration of the density around $h$.

However, the system actually works with the full posterior distribution over $\theta_\rho$, not merely its expectation. Fig. 2 (middle-right) illustrate how the posterior evolves depending on the number of times that a program has been refined, showing that it initially concentrates its mass around $h(\rho)$, and shifts downward with each refinement, making it progressively less likely to be refined further, but maintaining the property that every program always has a nonzero chance of being the next refinement. The variance also decays following $\mathrm{Var}\left[\theta_\rho \mid N_\rho\right] = \mathcal{O}\left(N_\rho^{-3}\right)$, meaning that a program that has already been heavily refined is not only lower in expectation, but also less likely to be randomly 'bumped up' and promoted as the next arm to pull by Thompson Sampling.

## 5 Experimental Results

We study three different domains that each involve code generation. See Fig. 3. (Followup work [12] also applies *REx* to learning world models, where its scales to synthesizing 200+ line programs):

1. **Competition Programming.** We benchmark on APPS, one of the most challenging LLM programming problem benchmarks [13]. APPS problems involved generating self-contained Python code that solves an algorithmic puzzle, given a natural-language problem description. APPS is split into three difficulty levels: Competition, Interview, and Introductory. Competition-level problems are very challenging, with landmark LLMs such as AlphaCode only solving 8% of APPS Competition [14], making APPS substantially more difficult than basic benchmarks such as HumanEval [15] and MBPP [16].

2. **Visual Reasoning.** We take visual reasoning puzzles from the Abstraction and Reasoning Corpus (ARC [7, 17]). In ARC the goal is to infer an image-manipulating program from input-output examples, effectively combining inductive reasoning and visual/geometric puzzle solving. We work with a 40-problem subset of ARC introduced in [18] that has been annotated with natural-language descriptions of the target function.

3. **Loop Invariant Synthesis for Software Verification.** Discovering loop invariants is an important formal verification task; see Appendix A.3 for a primer on this problem statement. We collect 38 *non-linear* loop invariant synthesis tasks [19] from [20, 21]. These previous works rely on manually supplied templates/features and on dynamic program analyses (i.e., running the code). In contrast, we solely use source code without any dynamic execution traces or feature engineering. We check all three criteria of being a sufficiently strong inductive invariant (i.e., precondition, induction, and postcondition) with the Z3 solver [22]. The Z3 solver timeout is set to 2 minutes.

We use these datasets to study several research questions: (1) For a given compute budget, which approaches to refinement solve the most problems? (2) Can *REx* solve hard problems other methods cannot? (3) How much of a speedup/cost-reduction does *REx* grant? (4) How sensitive are different methods to hyperparameter choices, and what are reasonable default hyperparameter settings for this new method? To investigate these questions, we study a range of baselines:

## Visual Reasoning
### Abstract and Reasoning Corpus

**Visual Reasoning Puzzle**

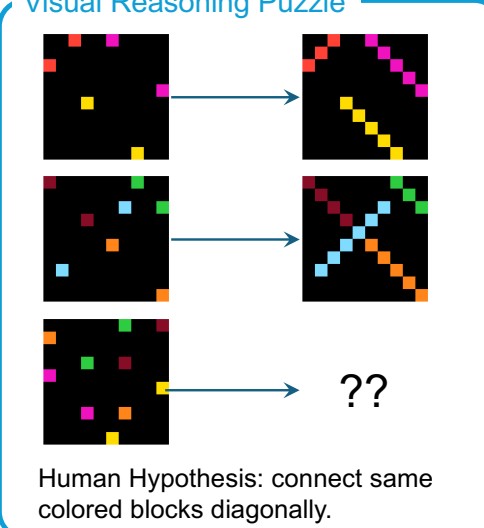

??

Human Hypothesis: connect same colored blocks diagonally.

**Image-Manipulating Program**

```python
def transform_grid(input_grid: np.ndarray) -> np.ndarray:
    color_list = np.unique(input_grid)[1:]
    for color in color_list:
        indices = np.argwhere(input_grid == color)
        if len(indices) > 1:
            y0, x0 = indices[0]
            for i in range(1, len(indices)):
                y1, x1 = indices[i]
                steps = max(abs(y1 - y0), abs(x1 - x0))
                steps = max(steps, 1)
                dy = (y1 - y0) / steps
                dx = (x1 - x0) / steps
                for s in range(1, steps):
                    y = round(y0 + dy * s)
                    x = round(x0 + dx * s)
                    input_grid[y][x] = color
                y0, x0 = y1, x1
    return input_grid
```

## Software Verification
### Nonlinear Loop Invariants

**Code to Verify**

```c
int mainQ(int x, int y){
    assert(x >= 1);
    assert(y >= 1);
    int a=x,b=y,p=1,q=0,r=0,s=1;
    while(1){
        //Loop Invariant in Z3:
        if(a==b) break;
        if (a>b) {
            a = a-b;
            p = p-q;
            r = r-s;
        }
        else {
            b = b-a;
            q = q-p;
            s = s-r;}
    }
    //assert(x%a==0 && y%a==0)
    //assert(∀n > a, x%n ≠ 0 || y%n ≠ 0)
    return a;
}
```

**Loop Invariant**
(Logical properties always hold during all iterations)

$$p * x + r * y == a$$
$$\&\& \ q * x + s * y == b$$
$$\&\& \ (p * s - q * r == 1$$
$$\| \ p * s - q * r == -1)$$
$$\&\& \ a \geq 0 \ \&\& \ b \geq 0$$

## Competition Programming, APPS

**Programming Problem**

Theta has been learning about powers of 2 in school. She notices that some numbers when written out contain powers of 2 in their digit representation: for instance, 12560 contains 256 which is a power of 2. She has been wondering how many such numbers there are.

Can you write a program that counts how many numbers contain a given power of 2?

**Python Code**

```python
def solve(n, e):
    power2 = str(2 ** e)
    p = len(power2)
    n = str(n)
    n = '0' * (p - 1) + n
    dp = [[[0 for _ in range(p + 2)]
           for _ in range(2)] for _ in range(len(n) + 1)]
    dp[0][0][0] = 1

    # start a new potential match from a middle part of power of 2
    next_power2 = [0] * (p + 1)
    for i in range(1, p):
        j = next_power2[i - 1]
        while j > 0 and power2[i] != power2[j]:
            j = next_power2[j - 1]
        if power2[i] == power2[j]:
            j += 1
        next_power2[i] = j

    for i in range(len(n)):
        for j in range(2):
            for k in range(p + 1):
                d_range = (10 if j else int(n[i]) + 1)
                for d in range(d_range):
                    nj = j or d < int(n[i])
                    nk = k
                    if k < p:
                        while nk > 0 and str(d) != power2[nk]:
                            nk = next_power2[nk - 1]
                        if str(d) == power2[nk]:
                            nk += 1
                    dp[i + 1][nj][nk] += dp[i][j][k]

    return dp[len(n)][0][p] + dp[len(n)][1][p]
```

Figure 3: Evaluation domains. For visual reasoning, the goal is to synthesize an image-manipulating program that translates input images to the correct outputs. For software verification, the goal is to synthesize logical conditions as a valid loop invariant, in order to formally verify the functionality of the code. For competition programming, the goal is to generate an algorithm in Python.

1. **Greedy baseline** refines the program with the highest heuristic value. It has a hyperparameter corresponding to the heuristic value of the initial (blank) program.

2. **Breadth-First Search** expands the refinement tree breadth-first. Its single hyperparameter is the width (branching factor), searching deeper and wider the larger the compute budget.

3. **Fixed-Width** generates a fixed number of initial programs, and then refines them round-robin, while never refining the same program more than once [23, 24]. This generates a refinement tree of fixed width, searching deeper (but not wider) the larger the compute budget. There is a single hyperparameter corresponding to the initial width.

**Problems solved as a function of compute budget.**  Fig. 4 and Fig. 5 show programming problems solved as the number of refinements increases, varying method and hyperparameters. We consider both the number of problems solved at each level of compute budget, as well as the total Area Under Curve (AUC). While the exact rank order of methods depends on the dataset and the LLM, we see consistently that *REx* solves the most problems. However, the final number of problems solved is typically only modestly larger for *REx*. What method counts as second-best varies by domain. For example, Fixed-Width is nearly as good as *REx*  on APPS-Competition and ARC, but on Loop Invariants and APPS-Introductory, Fixed-Width is the worst-performing method.

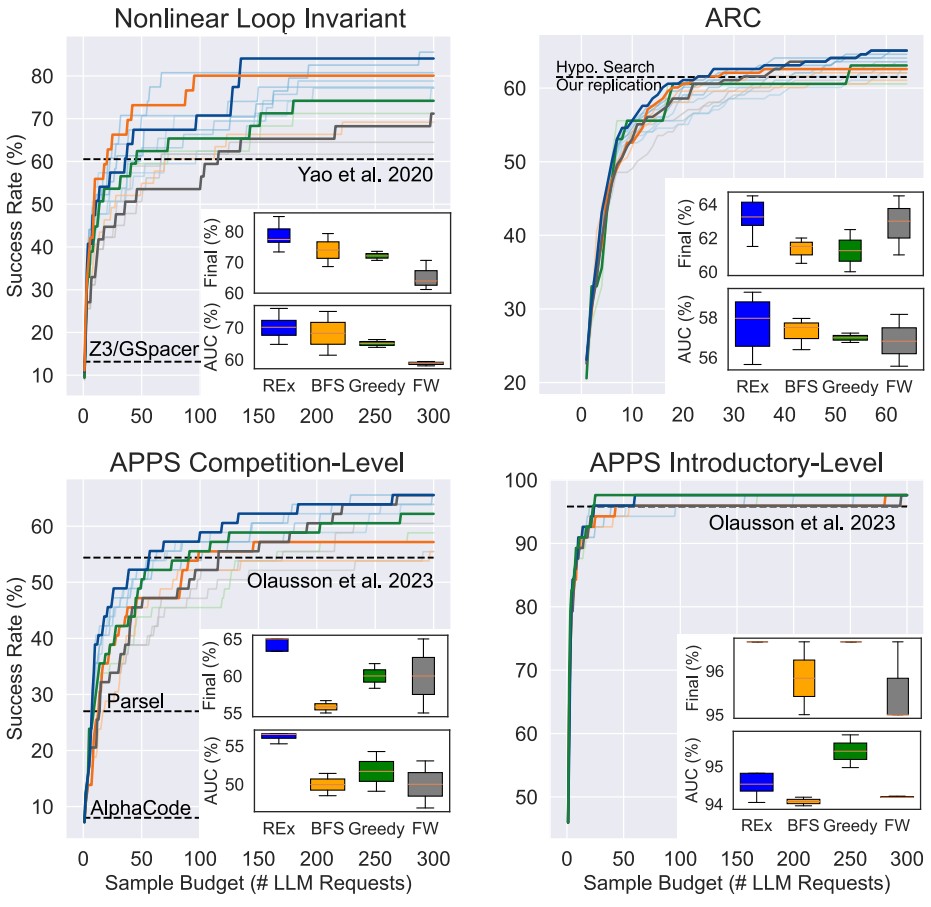

Figure 4: Comparing *REx* with alternatives using GPT-4 (temp=1). BFS and FW are Breadth First Search and Fixed Width, respectively. AUC denotes Area Under the Curve, and Final denotes the success rate at the maximum # LLM calls (64 for ARC and 300 for others due to domain conventions). Dark lines show performance with the best hyper-parameter setting for each method. Light lines show each run on each hyperparameter. The inset box plots show the distribution while varying the hyper-parameters. APPS baselines: Parsel [25], AlphaCode [14], and Olausson et al. [3]. Nonlinear Loop Invariant baselines: Z3/GSpacer [26] and Yao et al. [21]. ARC baseline: Hypo. Search [23]. More results on APPS Interview-Level and ARC in Figure 11 and Figure 13

We hypothesize that our robustness across datasets may come from the ability of *REx* to adaptively choose whether to search wide or deep. The other methods commit to a particular depth/width tradeoff, and how to optimally balance depth/width can vary by dataset. Indeed, examining the trees produced by *REx* shows that the algorithm searches quite broadly but refines deeply along more promising branches (Fig. 7), in ways that are reminiscent of Monte Carlo Tree Search.

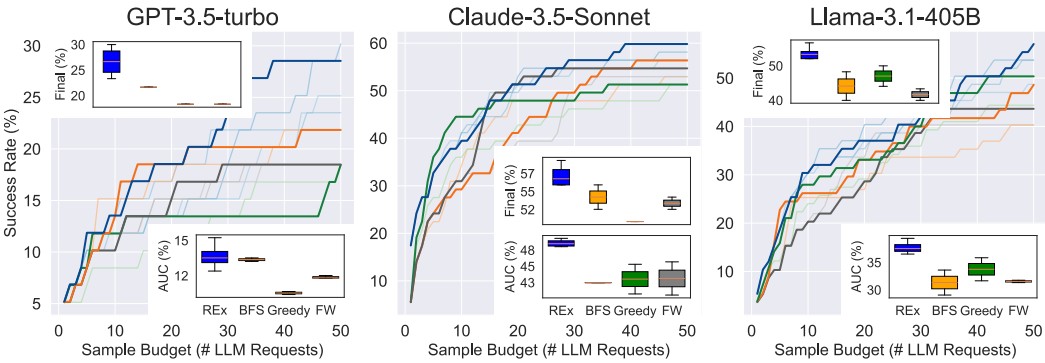

Figure 5: Comparing *REx* with alternatives with other LLMs on competition programming (APPS Competition-Level). More results on ARC are available in Appendix in Figure 12.

**Speedups.** Given the high monetary and environmental cost of frontier LLMs, it is valuable to understand whether different methods demand different levels of compute when solving the same problems. Fig. 6 analyzes these speedups, finding that *REx* requires less compute than alternatives: roughly $2\times$ less on APPS-Competition, $2$-$5\times$ less on loop invariants, and $1.1$-$1.6\times$ less on ARC. We also see that there is little advantage to our method on easier benchmarks: for example, there is no speedup on APPS Introductory. By definition, harder problems take more time, therefore results on the hardest problems are the most relevant for saving time and compute.

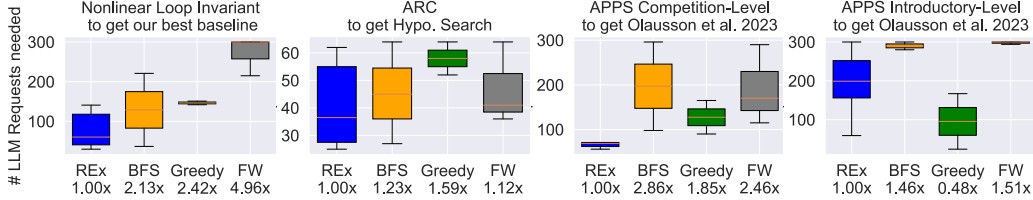

Figure 6: Number of LLM Requests needed to get the best LLM-based state-of-the-art methods (for ARC, APPS) and the best LLM-based baseline (for Nonlinear Loop Invariant). The lower the better. Box plot shows range of results across hyperparameters. "x" denotes the factor by which our method is faster than the compared baseline on this benchmark (ratio of median #LLM calls).

**Solving hard problems.** In the large compute limit *REx* asymptotically solves modestly more problems than the baselines, particularly on harder datasets (Fig. 4, APPS Competition vs Intro). How do these performance levels compare to other recent works on the same problem domains? On ARC, the state-of-the-art is Hypothesis Search [23], which we replicate, finding *REx* solves more problems.[1] On Loop Invariants, we solve more problems than general purpose solvers such as Z3-GSpacer [26, 22] as well as the state-of-the-art solver that is specially designed for this dataset [21]. The specially designed solver individually tuned the feature space and the hyper-parameters for each benchmark problem. It also assumes access to runtime traces of the code, and the ability to run the code thousands of times. Yet *REx* outperforms it by a large margin (73.7% v.s. 60.5%[2]), with no special tuning. *REx* is also, to the best of our knowledge, state-of-the-art on APPS-Competition with GPT4 (Fig. 4). *REx* therefore not only solves problems that our baselines cannot, but can set new state of the arts on the specific datasets we apply it to.

---

[1] Our replication gets higher numbers than [23], potentially due to using different subsets of ARC.

[2] Our evaluation also checks if the invariant is valid (Fig. 8), so the number is lower than reported in [21].

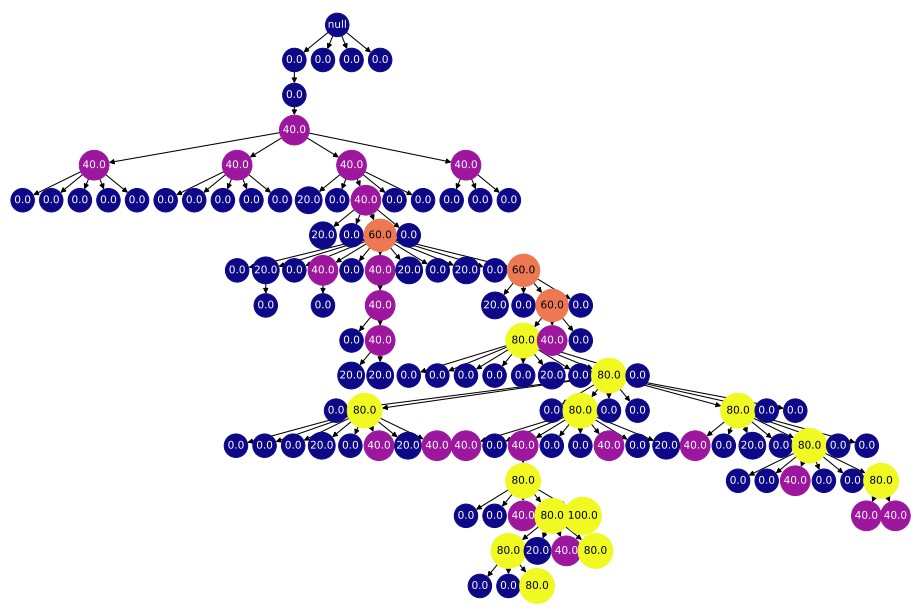

Figure 7: Example search tree generated by *REx*. Blue→Yellow gradient indicates heuristic value (blue: low, yellow: high). The order of children from left to right indicates the sequence of generations (left: older, right: newer). See also appendix Fig. 14-17

**Hyperparameter sensitivity.** We consider a range of hyperparameters for each method. Box-and-whisker plots in Fig. 4, Fig. 5, and Fig. 6 illustrate the distribution of results for different methods, while varying hyperparameters. For *REx*, large $C$ values work well on all datasets ($C = 20$). For the other methods, the optimal hyperparameter varies more widely: For example, even though Olausson et al. [3] suggests larger width for the fixed-width approach, we find smaller width can be better, depending on the dataset: Fixed-Width benefits from larger widths on ARC and Nonlinear Loop Inv, but performs better on APPS at smaller widths. Appendix Tbl. 1 shows these and other results in more detail, finding that our method is significantly more robust to variation in hyperparameters, while the other methods degrade more when their hyperparameters are not tuned to each specific dataset. We speculate that this may be due to the same reason that *REx* is more robust across domains: that it can adaptively decide when and where to search wide or deep.

## 6 Related Work

**Code refinement.** Prompting an LLM to fix problems with code that it generates has been explored in recent work [1, 4, 3]. To the best of our knowledge, our work is the first to systematically explore different policies for guiding refinement, and to introduce new bandit-based policies. Other works have explored fine-tuning specialized models to perform refinement, which allows using smaller models [6, 27], and which could be synergistic with our work here. More broadly within computer science, there is a long history of research on automated program repair using non-neural methods such as genetic search and constraint solving[28, 29, 30, 31, i.a.], which generally considers a different problem setting: assisting the fixing of human-written code, often within the context of a large code base. It is also possible to combine these classical methods with contemporary machine learning [32, 33].

**Bandits and Tree Search.** That tree search algorithms introduce an explore-exploit tradeoff has been long appreciated. Monte Carlo Tree Search (MCTS) is the canonical tree search algorithm that relies on this insight [34]. Although *REx* derives from a similar insight, its structure is very different from MCTS. Unlike MCTS, we do not perform rollouts, backups, or tree traversals. Instead, *REx* exploits unique properties of the refinement process: Any node can be further expanded (infinitely),

every node can be treated as a terminal node, and node expansions are very expensive (so rollouts would not be practical). To the best of our knowledge it is not possible to apply MCTS or any of its standard variants [35, 36, 37] to our problem statement out-of-the-box because of these unique properties. Bandits-based algorithms have also been applied for fuzz testing [38, 39] where code coverage is more important and code mutation is performed instead of code refinement.

## 7 Limitations and Future Directions

We see that our method is only modestly more effective at actually solving more problems overall in the large-compute limit: Its advantages are largest when viewed through the lens of minimizing cost, and of being robust across hyperparameters and datasets. Progress on cracking the hardest problems, such as solving all of ARC or all of APPS, is more likely to come from improvements to the base model, and less likely to come from improved ways of organizing the refinement search process.

We merely use the pass-rates of the programs to guide the search. There are richer information and more advanced bandits-based algorithms to integrate for better performance. A natural extension of our REx is to use contextualized bandits algorithms conditioned on the problem statement, the program, and the refinement history.

**Acknowledgements.** This work was supported by NSF grant #2310350, a gift from Cisco, and in part, by Individual Discovery Grants from the Natural Sciences and Engineering Research Council of Canada, and the Canada CIFAR AI Chair Program. We thank the Microsoft Accelerating Foundation Models Research (AFMR) program for providing generous support of Azure credits.

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

# A  Appendix

## A.1  Pseudocode

---

**Algorithm 1** Bandit formulation of program synthesis

---

**Input:** logical constraint $\Phi$, heuristic $h(\cdot)$, seed program $\rho$ (such as an empty blank function)
**Hyperparameter:** $C > 0$
   $\texttt{progs} \leftarrow \{\rho\}$             $\triangleright$ initialize with one arm (one program)
   $N \leftarrow \texttt{dict()}$     $\triangleright$ counter for each program tracking how many times it has been refined
   $N_\rho \leftarrow 0$             $\triangleright$ initially, no refinements
   **repeat**
   $\triangleright$ Thompson Sampling
      $\forall \rho \in \texttt{progs} : \theta_\rho \sim \mathrm{Beta}(1 + C \times h(\rho), 1 + C \times (1 - h(\rho)) + N_\rho)$     $\triangleright$ sample
      $\rho \leftarrow \arg\max_{\rho \in \texttt{progs}} \theta_\rho$     $\triangleright$ select arm
      $\rho' \sim P_{\mathrm{refine}}(\cdot | \rho, \Phi)$     $\triangleright$ call LLM to refine
      $N_\rho \leftarrow N_\rho + 1$     $\triangleright$ increment count of number of refinements
   $\triangleright$ Arm arrival
      $\texttt{progs} \leftarrow \texttt{progs} \cup \{\rho'\}$     $\triangleright$ add to set of arms
      $N_{\rho'} \leftarrow 0$     $\triangleright$ initialize counter to zero
   **until** $\rho' \vdash \Phi$
   **return** $\rho'$

---

## A.2  Hyperparameter analysis

Table 1: Analyze the hyperparameter sensitivity of methods by their performance rankings on different benchmarks. The rankings are determined by (AUC + final_success_rate) / 2. The hyperparameter, empty value for Greedy, denotes the heuristic value assigned for the empty solution. The hyperparameter choices for ARC are different from the others because of its domain convention (maximum # LLM requests=64 instead of 300). REx consistently outperforms or competes with the best baselines when $C = 20$ for all difficult benchmarks.

| | LoopInv | ARC | APPS-Comp. | APPS-Inter. | APPS-Intro. |
|---|---|---|---|---|---|
| Greedy (empty value=0) | 6 | 9 | 7 | 1 | 1 |
| Greedy (empty value=0.5) | 4 | 13 | 10 | 8 | 3 |
| BFS (branching factor=2) | | 12 | | | |
| BFS (branching factor=3) | 8 | 10 | 9 | 11 | 8 |
| BFS (branching factor=4) | | 7 | | | |
| BFS (branching factor=5) | 7 | | 12 | 8 | 12 |
| Fixed-Width (width=2) | | 14 | | | |
| Fixed-Width (width=4) | | 8 | | | |
| Fixed-Width (width=8) | | 4 | | | |
| Fixed-Width (width=25) | 12 | | 6 | 3 | 7 |
| Fixed-Width (width=50) | 10 | | 8 | 5 | 10 |
| Fixed-Width (width=100) | 11 | | 12 | 10 | 9 |
| REx (C=5) | 5 | 11 | 5 | 9 | 5 |
| REx (C=10) | 9 | 5 | 4 | 7 | 2 |
| REx (C=15) | | 3 | | | |
| REx (C=20) | 3 | 2 | 1 | 4 | 11 |
| REx (C=25) | | 1 | | | |
| REx (C=30) | | 6 | | | |
| REx (C=50) | 2 | | 2 | 6 | 4 |
| REx (C=100) | 1 | | 3 | 2 | 6 |

### A.3 Loop Invariants

**Solving for loop invariants** is a core problem in the program verification domain. For each program, we can write down its required specifications as logical constrains on its inputs and outputs. One can then statically step through the program execution to check if the minimal preconditions on the inputs can ensure the post conditions on the outputs.

However, *loops* in the program have dynamic repetitions, which cannot be directly described under Satisfiability Modulo Theories (SMT) checkers. This raises the question to find strong invariant conditions about the loop, which should summarize the loop's behaviour and allow the SMT checker to reason on the preserved properties of the variables.

$$pre \Rightarrow inv, \quad (inv \wedge lc \wedge trans) \Rightarrow inv', \quad (inv \wedge \neg lc) \Rightarrow post$$

Figure 8: Verification conditions of a loop: *pre* - preconditions, *lc* - loop condition, *trans* - state transition function, $\neg lc$ - loop termination condition, *post* - post conditions

$$
\begin{aligned}
x &:= -50; \\
\textbf{while } &(x < 0) \{ \\
x &:= x + y; \\
y &:= y + 1 \} \\
\textbf{assert} &(y > 0)
\end{aligned}
$$

(a) An example program.

(b) A desirable loop invariant $I$ is a predicate over $x, y$ such that:

$$\forall x, y : \begin{cases} \texttt{true} \Rightarrow I[-50/x] & (pre) \\ I \wedge x < 0 \Rightarrow I[(y+1)/y, (x+y)/x] & (inv) \\ I \wedge x \geq 0 \Rightarrow y > 0 & (post) \end{cases}$$

(c) The desired loop invariant is $(x < 0 \vee y > 0)$.

Figure 9: A concrete example of a loop invariant taken verbatim from Si et al. [40]. Loop invariant connects the pre-conditions with the post-conditions to prove, using logical properties that always hold during the iterations. "/" denotes the value assignment. $I[-50/x]$ means substituting $x$ in the loop invariant, $I$, with $-50$. More details regarding loop invariant are available in Si et al. [40].

The specifications and the loop behaviour together describe the **verification conditions** (VC) for the program. Finding an invariant statement that satisfies these requirements proves the correctness of the loop's functionality. Specifically, the invariant is *established* if it can be deduced from the preconditions; it is *preserved* if it maintains itself after each loop iteration; and, it is *strong* if leading to the post conditions.

Existing methods often fit for invariants by running program traces or by guessing heuristics based on the feedback from the checker. In this paper, we incorporate the large language model's code understanding and reasoning ability to analyze a program's behavior and to produce sufficient loop invariants.

#### A.3.1 NLA Benchmark Dataset

The verification of some programs require a particular subset of loop invariants that contain *non-linear polynomial arithmetic* (NLA) terms with regards to the program's variables. Finding NLA invariants is not well addressed by state-of-the-art formal solvers. Nevertheless, LLMs can produce arbitrary formula and are not constrained by the forms of the invariants. Thus, utilizing these models is a promising approach to address the performance gap and to discover novel invariants [41].

Here, we present a set of programs that require NLA invariants to prove their functional correctness [42]. All of these programs are meaningful algorithms on arithmetics that have real-world applications and have (multiple) non-linear loops. Examples include sums of powers, extended Euclidean algorithm to calculate greatest common divisors, and binary division algorithm to speedup calculation in hardware.

Each instance in the benchmark contains the source code in C and the verification conditions for each of its loops. For programs with multiple loops, at least one loop requires NLA invariants. We manually verify the VCs and provide a ground truth reference invariant for each loop. We write the VCs and the reference invariants in the Z3Py format. We then use the **Z3** theorem prover to check each invariant against the VCs. This way, we can statically verify whether the program's behaviour satisfy the specifications.

| Index | Program | Loop # | InvType | G-CLN [21] | GSpacer [26] | Loopy-GPT4 [41] | **REx** |
|---|---|---|---|---|---|---|---|
| 1 | cohencu | 1 | NL | ✓ | - | - | - |
| 2 | cohendiv | 1 | NL | ✓ | - | ✓ | ✓ |
| 3 | | 2 | NL | - | - | ✓ | ✓ |
| 4 | dijkstra | 1 | Linear | ✓ | ✓ | - | ✓ |
| 5 | | 2 | NL | ✓ | - | - | - |
| 6 | divbin | 1 | Linear | ✓ | ✓ | - | ✓ |
| 7 | | 2 | NL | ✓ | - | ✓ | ✓ |
| 8 | egcd | 1 | NL | ✓ | - | - | ✓ |
| 9 | egcd2 | 1 | NL | - | - | - | ✓ |
| 10 | | 2 | NL | - | - | ✓ | ✓ |
| 11 | egcd3 | 1 | NL | - | - | - | ✓ |
| 12 | | 2 | NL | - | - | ✓ | ✓ |
| 13 | | 3 | NL | - | - | - | ✓ |
| 14 | fermat1 | 1 | NL | ✓ | - | - | ✓ |
| 15 | | 2 | Linear | - | - | - | ✓ |
| 16 | | 3 | Linear | - | - | ✓ | ✓ |
| 17 | fermat2 | 1 | NL | ✓ | - | - | ✓ |
| 18 | freire1 | 1 | NL | - | - | - | - |
| 19 | freire2 | 1 | NL | ✓ | - | - | ✓ |
| 20 | geo1 | 1 | NL | ✓ | - | - | - |
| 21 | geo2 | 1 | NL | ✓ | - | - | - |
| 22 | geo3 | 1 | NL | ✓ | ✓ | - | - |
| 23 | hard | 1 | NL | ✓ | - | ✓ | ✓ |
| 24 | | 2 | NL | ✓ | - | ✓ | ✓ |
| 25 | knuth | 1 | NL | - | - | - | - |
| 26 | lcm1 | 1 | NL | - | - | - | ✓ |
| 27 | | 2 | NL | ✓ | ✓ | - | ✓ |
| 28 | | 3 | NL | ✓ | ✓ | - | ✓ |
| 29 | lcm2 | 1 | NL | - | - | - | - |
| 30 | mannadiv | 1 | NL | ✓ | - | ✓ | ✓ |
| 31 | prod4br | 1 | NL | - | - | - | ✓ |
| 32 | prodbin | 1 | NL | ✓ | - | ✓ | ✓ |
| 33 | ps2 | 1 | NL | ✓ | - | ✓ | ✓ |
| 34 | ps3 | 1 | NL | ✓ | - | - | ✓ |
| 35 | ps4 | 1 | NL | ✓ | - | - | ✓ |
| 36 | ps5 | 1 | NL | - | - | - | - |
| 37 | ps6 | 1 | NL | ✓ | - | - | - |
| 38 | sqrt | 1 | NL | ✓ | - | - | ✓ |
| | Total correct | | | 24 | 5 | 11 | 28 |

Table 2: Benchmark evaluation on the NLA programs. For programs with multiple loops, each loop is labeled with a number, top to down, from outer to inner.

### A.3.2 LLM Invariant Generation

We generate potential invariants with GPT-4 by feeding in the C program and asking for Z3Py invariants. To identify the targeted loop, we mark the start of the loop with a comment, where the LLM should analyze the semantics on.

We use Z3's simplification function to split the invariants into conjunctive normal form and check each item against the VCs. When an item from the CNF invariants does not satisfy all steps in the checking process, we filter out the candidate using the Houdini pruning algorithm, which is commonly used in invariant inference [41]. The final result is evaluated after performing Houdini.

In the refinement setting, we use feedback from the Z3 checker to tell the LLM which step of the check failed, so that it knows whether a stronger or a more general invariant is needed. More detailed examples are shown in the prompts in Appendix A.6.1.

### A.3.3 Baseline Methods

**G-CLN** is a learning-based method that fits for invariants by running program traces [21] and achieved the state-of-the-art result on the NLA benchmark. Although it is designed to perform polynomial fitting on the program variables, it relies on manually crafted features that are instance-specific heuristics (i.e., each instance requires a different set of features). In addition, this dynamic approach requires sampling different program runs, which can omit corner cases, and is unable to handle programs with non-deterministic termination condition. Our method and many others avoid this problem using the static analysis paradigm.

Examples where G-CLN fails on the NLA benchmark can be summarized in several reasons:

- The loop requires high order invariants beyond the specified heuristic, as in *knuth*

- The method fits on and generates excessive unnecessary weak invariants that inhibit the checker, as is the case in *ps5*

- The method finds an invariant based on the global traces but is too strong on the current loop

- Unable to handle edge case logic involving floating point calculation

- Unable to prove the correctness of the Greatest-Common-Divisor algorithms when having no access to numpy.gcd.

**Spacer** is Z3's builtin fixedpoint engine. It can look for inductive invariants for a loop after giving it the VC definitions, formulated in Constrained Horn Clauses (CHCs). We test on the newest version called GSpacer that incorporates global guidance during solving[26], which is the state-of-the-art CHC solver. Although GSpacer can effectively solve linear invariants using global insights, where previous myopic heuristics were to cause non-terminating runs, we observe that it is only able to solve 5 problems from the NLA benchmark, with 3 of them requiring actual non-linear polynomial invariants.

### A.3.4 Summary of Program Types

| Problem Type | Programs |
|---|---|
| Integer Division Algorithms | divbin, mannadiv, cohendiv, hard |
| Integer Power Algorithms | cohencu |
| GCD & LCM | egcd, egcd2, egcd3, lcm1, lcm2 |
| Divisor for Factorisation | fermat1, fermat2, knuth |
| Geometric Sequences | geo1, geo2, geo3 |
| Integer Multiplication | prodbin, prod4br |
| Square Root Calculation | sqrt, dijkstra, freire1, freire2 |
| Power of Sums | ps2, ps3, ps4, ps5, ps6 |

Table 3: Categorized NLA problems

## A.4 The Abstraction and Reasoning Corpus

### A.4.1 Dataset

The Abstraction and Reasoning Corpus(ARC) is a visual reasoning task proposed by Chollet[7]. It contains 400 training and 400 evaluating problems. Each problem will provide the task taker with several input-output images expressed by 2D metrics pairs that follow the same transformation rule, such as rotating, expanding, and color-changing. The task taker is then asked to summarize the rule and give the correct output metric of the test input. Figure 10 shows two examples of the problem of the ARC dataset. The size of the metrics ranges from $1 \times 1$ to $30 \times 30$, and there are only 10 colors for each pixel represented by number 0 to number 9. In our paper, we use the subset of ARC proposed by Johnson[18], which contains 40 ARC problems with human experiments. We select one hypothesis written by human beings for each problem as a hint for LLMs.

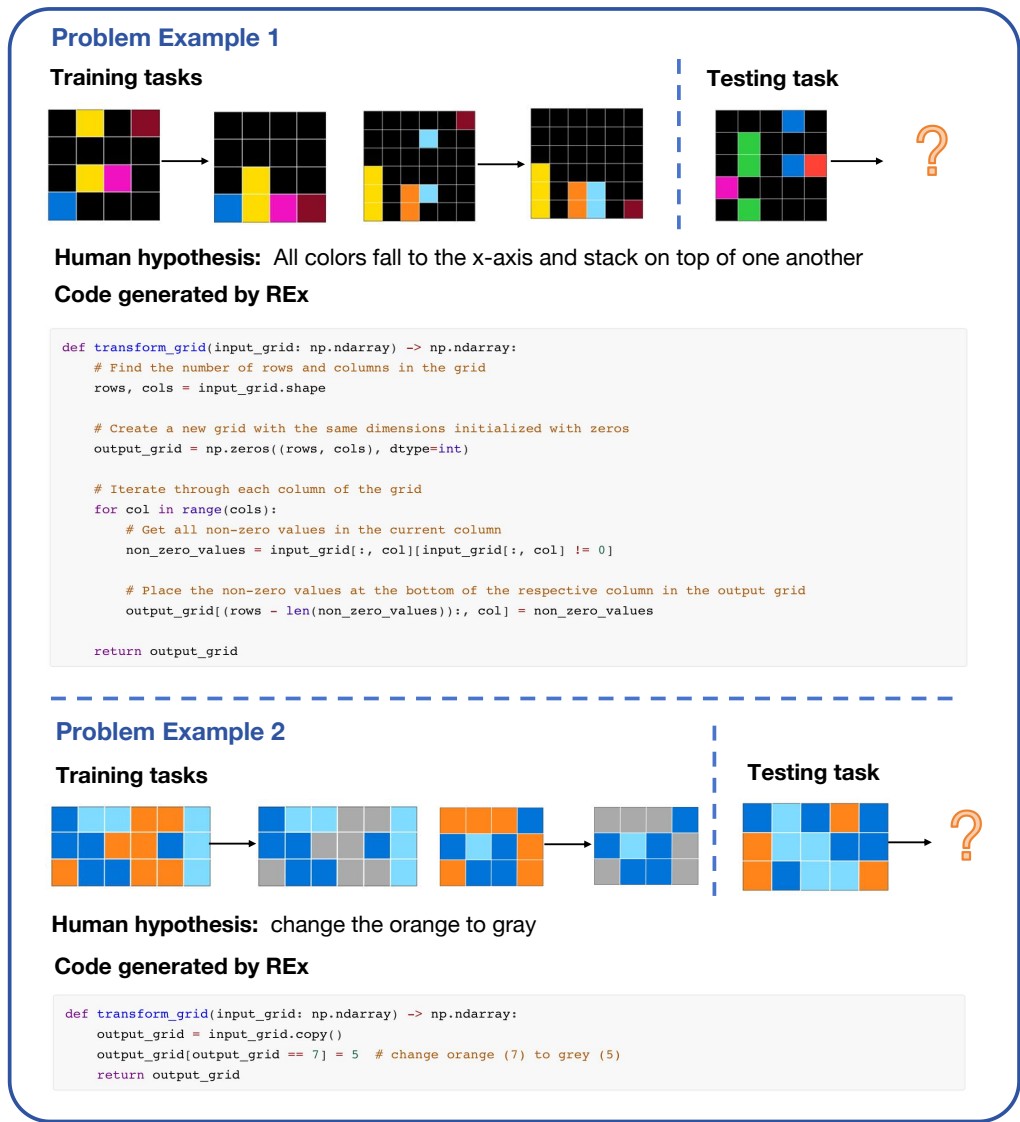

Figure 10: Problem examples of the ARC dataset.

### A.4.2 Baseline

We use the Human-Written Hypotheses experimental set in the paper hypothesis search[23], which gains the current state-of-art results of the ARC dataset, as our baseline. The paper uses a Fixed-Width searching policy to get the candidate programs, and it sets the width into 8 and the depth into 3. We run the baseline with the same searching policy and hyperparameter–a fixed width of 8 and its depth with 3, which accounts for maximum #LLM requests= 32. The prompts in our experiments are also consistent with the paper.

### A.4.3 Experimental setting

We use the subset of ARC, which contains 40 problems, as our dataset. Each problem contains several training tasks as examples. We utilize GPT-4 Turbo to generate code that summarizes the transformation rules and refine code that fails to pass the training examples. For each problem, we set REx's heuristic reward of each code as the pass rate of each problem, which means the percentage of correctly solved training tasks. If the code passes all the training examples, which means it

Table 4: Analyze the hyperparameter sensitivity of methods by their performance rankings on different benchmarks on the ARC domain with maximum #LLM requests=200. The ranking principle is the same as in Table 1.

| Methods | REx (C) | | | | Greedy | | BFS | | | Fixed-Width | | |
|---|---|---|---|---|---|---|---|---|---|---|---|---|
| hyperparameter type | C | | | | empty value | | branching factor | | | width | | |
| hyperparameter value | 10 | 25 | 50 | 100 | 0 | 0.5 | 2 | 4 | 16 | 10 | 25 | 50 |
| ranking | 5 | 2 | 1 | 9 | 6 | 3 | 8 | 10 | 12 | 11 | 4 | 7 |

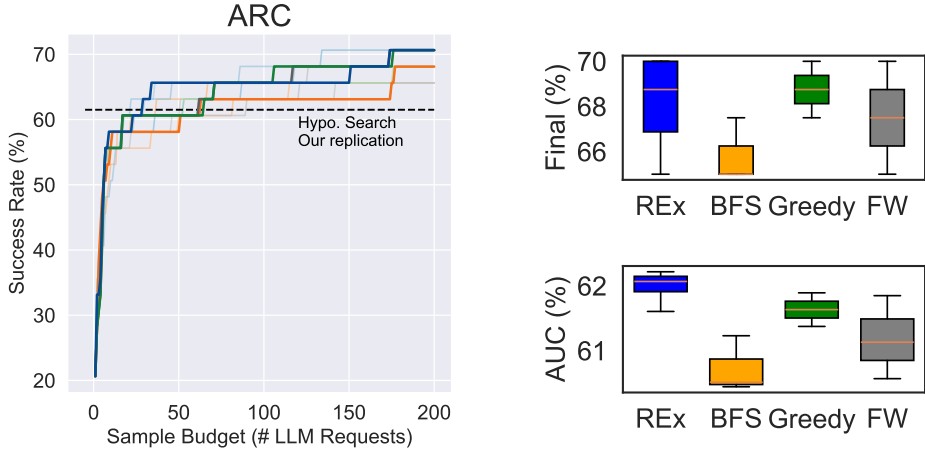

Figure 11: Results on the ARC domain with maximum #LLM requests=200. Denotations are the same as Figure 4.

successfully finds the candidate code that summarizes the transformation rule that can solve the problem, then the search process will stop.

We run our experiment with a maximum iteration of 64 for a fair comparison with Hypothesis Search [23]. We set the hyperparameters of each method accordingly as follows:

1. **Greedy:** empty value$= 0, 0.5$
2. **BFS:** branching factor$= 2, 3, 4$
3. **Fixed-Width:** width$= 2, 4, 8$
4. **REx:** C$=5, 10, 15, 20, 25, 30$

For each method and each hyperparameter, we run them five times with five random seeds and report the average performance.

### A.4.4   Scaled-up results with 200 LLM requests

We also scale up the maximum iteration to 200 and demonstrate the benefits of REx in this setting as well. The hyperparameters are as follows:

1. **Greedy:** empty value$= 0, 0.5$
2. **BFS:** branching factor$= 2, 4, 16$
3. **Fixed-Width:** width$= 10, 25, 50$
4. **REx:** C$=10, 25, 50, 100$

Results are shown in Figure 11 and Table 4. REx overall achieves better performance than the other methods on the scale of 200 as well. It uses fewer LLM requests to reach a certain passing rate and gains an overall higher final success rate. REx achieves relatively better performance when $C \geq 20$.

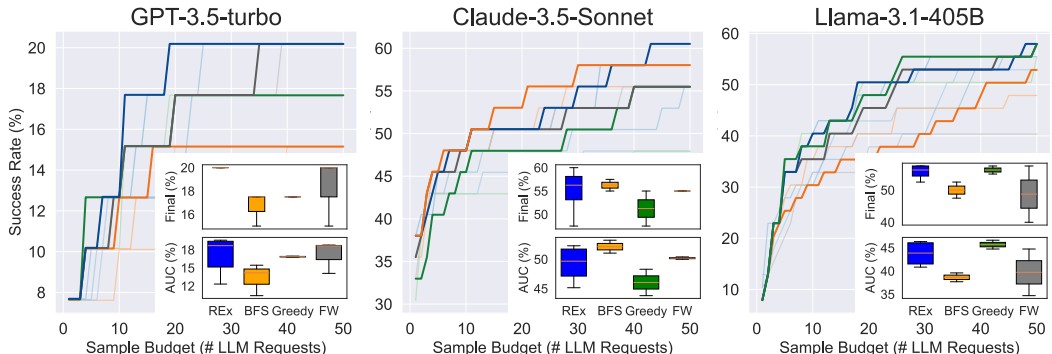

Figure 12: Comparing *REx* with alternatives with other LLMs on ARC: GPT-3.5-turbo, Claude-3.5-Sonnet, and Llama-3.1-405B.

## A.5 Competition Programming, APPS

### A.5.1 Dataset

We use the exact same subset of APPS as Olausson et al. [3] for fair comparison. We refer the reader to its Appendix H for details. We use all 60 problems for the competition-level and the introductory-level, and randomly choose 60 of 120 problems for the interview-level.

### A.5.2 Experimental setting

Most experiments are conducted with GPT-4 and temperature=1.0 by default. The choices of hyperparameters for each method are listed in Table 1. We also experiment with other LLMs including GPT-3.5-turbo, an older model to alleviate concerns regarding data contamination.

### A.5.3 Results on APPS-Interview

Figure 13 shows the results on APPS-Interview. As expected, REx performs competitively with the best baseline, Greedy, and outperforms the other two baselines. It consistently outperforms the state-of-the-art method, Olausson et al. [3], which uses the fixed-width method with width=25/50. It also aligns with our hypothesis that greedy performs competitively with REx when the dataset is relatively easy.

## A.6 Prompts

We list all the prompts that are used in our experiments in this section. The functionality for each prompt is stated in its subsection name. We highlight the dynamic information as yellow and the main instruction as blue. The dynamic information includes the codes synthesized so far by previous LLM calls.

### A.6.1 Loop Invariant Synthesis

**Initializing loop invariant.**    It asks LLMs to generate loop invariant to help understand the code and prove its correctness.

```
-----  Role: system  --------------------
You are an expert software engineer and you are working on a project that
requires you to write loop invariants for a loop in the code. The loop
invariant is a condition that must be true before and after each iteration
of the loop. The loop invariant helps you understand the behavior of the
loop and ensures that the loop is working correctly. The loop invariant
should be strong enough to prove the correctness of the loop as well as the
functionality of the whole code snippet.
```

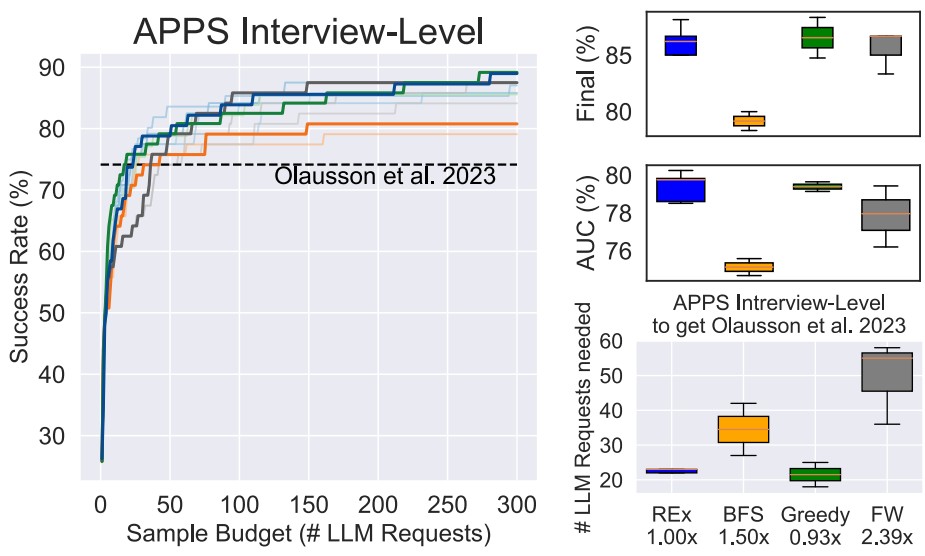

Figure 13: Results on APPS-Interview. Denotations are the same as Figure 4.

```
-----  Role: user  --------------------
You are given the following code snippet in C:

```

#include <stdio.h>

int main(int a, int b){
    assert(a>=1);
    assert(b>=1);
    int x,y,u,v;

    x=a;
    y=b;
    u=b;
    v=0;

    p=1;
    q=0;
    r=0;
    s=1;

    while(1) {
        // loop invariant in Z3:

        if (!(x!=y)) break;

        while (1){

            if(!(x>y)) break;
            x=x-y;
            v=v+u;

                p = p-q;
            r = r-s;
        }
```

```
        while (1){

            if(!(x<y)) break;
            y=y-x;
            u=u+v;

            q = q-p;
            s = s-r;
        }

    }

    int r = u+v;
    return r; //lcm
}
```

‘‘‘

Do you understand the code snippet? What function does the code snippet
perform? Can you explain the code snippet part by part? What is the
functionality of each part of the code snippet? What is the function of the
loop in the code snippet?

The loop that needs a loop invariant has been identified in the code snippet
 with a line starting with ‘// loop invariant in Z3: ‘. Do you understand
the function of the loop in the code snippet? Can you explain the function
of the loop in the code snippet? What is the purpose of the loop in the code
 snippet?

What variables are there in the loop in the code snippet? How are the
variables used and updated in the loop in the code snippet? What is the
relationship among the variables in the loop in the code snippet?

Please analyze the loop variant in your own language. The loop invariant
should be a condition that must be true before and after each iteration of
the loop. The loop invariant should help you understand the behavior of the
loop and ensure that the loop is working correctly. The loop invariant
should also be strong enough to prove the correctness of the loop as well as
 the functionality of the whole code snippet.

What are the loop invariants in code? The loop invariant should be written
in the format as Python-Z3 such as follows:

‘‘‘

And(x >= 0, x <= 10, Implies(x >= 0, a == 0), y == x * x)

‘‘‘
Please surround the loop invariant with triple backticks and write it in the
 format as Python-Z3.

**Refining loop invariant.**    It asks LLMs to refine loop invariant given the error messages. In detail,
the loop invariant should satisfy three conditions in order to be valid and sufficient:

- Established: meaning that loop invariant should be true given the preconditions, i.e., it should be true at the start of the loop.

- Preserved: meaning that loop invariant should also be true after each iteration of the loop.

- Sufficient: meaning that the loop invariant should be strong enough to help us prove the post conditions, i.e., the specifications/properties assigned by the users.

The symbolic checker utilizes SMT solvers to evaluate these three conditions individually and returns the error messages. If any of the conditions is not satisfied, it will return the error type, the specific assertion, and its model to help debug. LLMs then refine the loop invariant given that information. Note that, we deliberately omit the detailed information about post conditions to avoid label leakage. Our loop invariant synthesis system can then also be applied in scenarios where user specifications are not explicitly available in logical forms.

```
-----  Role: system  --------------------
You are an expert software engineer and you are working on a project that
requires you to write loop invariants for a loop in the code. The loop
invariant is a condition that must be true before and after each iteration
of the loop. The loop invariant helps you understand the behavior of the
loop and ensures that the loop is working correctly. The loop invariant
should be strong enough to prove the correctness of the loop as well as the
functionality of the whole code snippet.
-----  Role: user  -------------------
You are given the following code snippet in C:

'''

#include <stdio.h>

int main(int a, int b){
    assert(a>=1);
    assert(b>=1);
    int x,y,u,v;

    x=a;
    y=b;
    u=b;
    v=0;

    p=1;
    q=0;
    r=0;
    s=1;

    while(1) {
        // loop invariant in Z3:

        if (!(x!=y)) break;

        while (1){

            if(!(x>y)) break;
            x=x-y;
            v=v+u;

                p = p-q;
            r = r-s;
        }

        while (1){
```

```
            if(!(x<y)) break;
            y=y-x;
            u=u+v;

            q = q-p;
            s = s-r;
        }

    }

    int r = u+v;
    return r; //lcm
}
```

''' 

Do you understand the code snippet? What function does the code snippet perform? Can you explain the code snippet part by part? What is the functionality of each part of the code snippet? What is the function of the loop in the code snippet?

The loop that needs a loop invariant has been identified in the code snippet with a line starting with '// loop invariant in Z3: '. Do you understand the function of the loop in the code snippet? Can you explain the function of the loop in the code snippet? What is the purpose of the loop in the code snippet?

What variables are there in the loop in the code snippet? How are the variables used and updated in the loop in the code snippet? What is the relationship among the variables in the loop in the code snippet?

According to those analysis, could you refine the following loop invariants that you generated before?

```
And(
    a >= 1,
    b >= 1,
    x >= y,
    a*b == x*y
)
```

The previous loop invariant is wrong because:

  - 'Or(Not(x >= 0), u == a*x + b)' is neither established nor preserved, meaning it is not even true in the beginning of the loop and is neither true after each iteration of the loop. For example, we can set [p = 1, q = 0, y = 7, r = 0, v = 0, s = 1, x = 15, b = 7, a = 15, u = 7] to find a counterexample for establishing 'Or(Not(x >= 0), u == a*x + b)', since it conflicts with the assertion 'Implies(And(a > 0, b > 0, x == a, y == b, u == b, v == 0, p == 1, q == 0, r == 0, s == 1), Or(Not(x >= 0), u == a*x + b))'.
  - 'Or(Not(y >= 0), v == a + b*y)' is neither established nor preserved, meaning it is not even true in the beginning of the loop and is neither true after each iteration of the loop. For example, we can set [v = 0, s = 1, p
```

```
= 1, x = 16, q = 0, u = 13, r = 0, y = 13, a = 16, b = 13] to find a
counterexample for establishing 'Or(Not(y >= 0), v == a + b*y)', since it
conflicts with the assertion 'Implies(And(a > 0, b > 0, x == a, y == b, u ==
 b, v == 0, p == 1, q == 0, r == 0, s == 1), Or(Not(y >= 0), v == a + b*y))
'.
  - The metrics are neither enough to imply the postconditions.

Please correct the previous loop invariants and provide the correct loop
invariants for the loop in the code snippet. The loop invariant should be
written in the format as Python-Z3 as before. Please surround the loop
invariant with triple backticks and write it in the format as Python-Z3.
```

### A.6.2   Code Generation

**Initializing code.**   It asks LLMs to generate code to solve python programming problems.

```
-----   Role: system   --------------------
You are a helpful assistant that can solve python programming problems and
debug incorrect programs
-----   Role: user   --------------------
You are given a hard python programming contest problem:

Question:

You are holding a party. In preparation, you are making a drink by mixing
together three different types of fruit juice: Apple, Banana, and Carrot.
Let's name the juices A, B and C.

You want to decide what fraction of the drink should be made from each type
of juice, in such a way that the maximum possible number of people attending
 the party like it.

Each person has a minimum fraction of each of the 3 juices they would like
to have in the drink. They will only like the drink if the fraction of each
of the 3 juices in the drink is greater or equal to their minimum fraction
for that juice.

Determine the maximum number of people that you can satisfy.

-----Input-----
 - One line containing an integer $T$, the number of test cases in the input
  file.

For each test case, there will be:
 - One line containing the integer $N$, the number of people going to the
party.
 - $N$ lines, one for each person, each containing three space-separated
numbers ''$A$ $B$ $C$'', indicating the minimum fraction of each juice that
would like in the drink. $A$, $B$ and $C$ are integers between $0$ and
$10000$ inclusive, indicating the fraction in parts-per-ten-thousand. $A + B
 + C \leq 10000$.

You may assume that $1 \leq T \leq 2$ and $1 \leq N \leq 5000$.

-----Output-----
 - $T$ lines, one for each test case in the order they occur in the input
file, each containing the string ''Case #$X$: $Y$'' where $X$ is the number
```

of the test case, starting from 1, and $Y$ is the maximum number of people
who will like your drink.

-----Examples-----
Sample Input:
2
3
10000 0 0
0 10000 0
0 0 10000
3
5000 0 0
0 2000 0
0 0 4000
Sample Output:
Case #1: 1
Case #2: 2

Can you write a correct solution and make sure it passes the example test
cases? (Please start the solution segment with ```python as usual and use
Standard Input format)

**Refining code.**   It asks LLMs to refine code given the test case it failed on and the error message.

```
-----  Role: system  -------------------
You are a helpful assistant that can solve python programming problems and
debug incorrect programs
-----  Role: user  -------------------
You are given a hard python programming contest problem:

Question:

You are holding a party. In preparation, you are making a drink by mixing
together three different types of fruit juice: Apple, Banana, and Carrot.
Let's name the juices A, B and C.

You want to decide what fraction of the drink should be made from each type
of juice, in such a way that the maximum possible number of people attending
 the party like it.

Each person has a minimum fraction of each of the 3 juices they would like
to have in the drink. They will only like the drink if the fraction of each
of the 3 juices in the drink is greater or equal to their minimum fraction
for that juice.

Determine the maximum number of people that you can satisfy.

-----Input-----
 - One line containing an integer $T$, the number of test cases in the input
  file.

For each test case, there will be:
 - One line containing the integer $N$, the number of people going to the
party.
 - $N$ lines, one for each person, each containing three space-separated
numbers ``$A$ $B$ $C$'', indicating the minimum fraction of each juice that
would like in the drink. $A$, $B$ and $C$ are integers between $0$ and
```

$10000$ inclusive, indicating the fraction in parts-per-ten-thousand. $A + B + C \leq 10000$.

You may assume that $1 \leq T \leq 2$ and $1 \leq N \leq 5000$.

-----Output-----
 - $T$ lines, one for each test case in the order they occur in the input file, each containing the string ``Case #$X$: $Y$'' where $X$ is the number of the test case, starting from 1, and $Y$ is the maximum number of people who will like your drink.

-----Examples-----
Sample Input:
2
3
10000 0 0
0 10000 0
0 0 10000
3
5000 0 0
0 2000 0
0 0 4000
Sample Output:
Case #1: 1
Case #2: 2

Below is a failed attempt to solve the problem:

```
t = int(input())
for case in range(1, t + 1):
    n = int(input())
    total_fractions = {'A': 0, 'B': 0, 'C': 0}
    for person in range(n):
        a, b, c = map(int, input().split())
        total_fractions['A'] += a
        total_fractions['B'] += b
        total_fractions['C'] += c
    scale = min(1, sum(value for value in total_fractions.values()) / 10000)
    scaled_fractions = {key: value * scale for key, value in total_fractions
.items()}
    count = 0
    for person in range(n):
        a, b, c = map(int, input().split())
        if a <= scaled_fractions['A'] and b <= scaled_fractions['B'] and c
<= scaled_fractions['C']:
            count += 1
    print('Case #{}: {}'.format(case, count))
```

The failure is due to runtime error at test case 0 for input 2\n3\n10000 0 0\n0 10000 0\n0 0 10000\n3\n5000 0 0\n0 2000 0\n0 0 4000\n and expected output Case #1: 1\nCase #2: 2\n. Error type: ValueError, detailed error message: not enough values to unpack (expected 3, got 1) at line 14, a, b, c = map(int, input().split())
Overall evaluation: 0 out of 2 test cases passed

Can you
1. Analyze why the runtime error occurs?
2. Correct the code and make sure it passes the example test cases? (Please start the solution segment with ```python as usual and use Standard Input format)

### A.6.3 ARC

**Initializing image-manipulating program.**  We ask LLMs to generate code that transforms the input matrices into the output matrices correctly, following the same transformation rule. The prompts are derived from Hypothesis Search[23].

```
-----  Role: user  --------------------
Example 0:
Input:
[[0 0 0 0 0 0 0 0 0]
 [0 6 6 6 0 0 0 0 0]
 [0 6 0 0 6 0 0 0 0]
 [0 0 6 0 0 6 0 0 0]
 [0 0 0 6 0 0 6 0 0]
 [0 0 0 0 6 6 6 0 0]
 [0 0 0 0 0 0 0 0 0]
 [0 0 2 2 2 0 0 0 0]
 [0 0 2 0 0 2 0 0 0]
 [0 0 0 2 2 2 0 0 0]
 [0 0 0 0 0 0 0 0 0]
 [0 0 0 0 0 0 0 0 0]
 [0 0 0 0 0 0 0 0 0]
 [0 0 0 0 0 0 0 0 0]]
Output:
[[0 0 0 0 0 0 0 0 0]
 [0 0 6 6 6 0 0 0 0]
 [0 0 6 0 0 6 0 0 0]
 [0 0 0 6 0 0 6 0 0]
 [0 0 0 0 6 0 6 0 0]
 [0 0 0 0 6 6 6 0 0]
 [0 0 0 0 0 0 0 0 0]
 [0 0 0 2 2 2 0 0 0]
 [0 0 0 2 0 2 0 0 0]
 [0 0 0 2 2 2 0 0 0]
 [0 0 0 0 0 0 0 0 0]
 [0 0 0 0 0 0 0 0 0]
 [0 0 0 0 0 0 0 0 0]
 [0 0 0 0 0 0 0 0 0]]
Example 1:
Input:
[[0 0 0 0 0 0 0 0 0]
 [0 8 8 8 8 8 0 0 0]
 [0 8 0 0 0 8 0 0 0]
 [0 0 8 0 0 0 8 0 0]
 [0 0 0 8 0 0 0 0 8]
 [0 0 0 0 8 8 8 8 8]
 [0 0 0 0 0 0 0 0 0]
 [0 0 0 0 0 0 0 0 0]]
Output:
[[0 0 0 0 0 0 0 0 0]
 [0 0 8 8 8 8 8 0 0]
 [0 0 8 0 0 0 0 8 0]
 [0 0 0 8 0 0 0 0 8]
 [0 0 0 0 8 0 0 0 8]
 [0 0 0 0 8 8 8 8 8]
 [0 0 0 0 0 0 0 0 0]
 [0 0 0 0 0 0 0 0 0]]

Now, please write a python program transform_grid(input_grid: np.ndarray[int
]) -> np.ndarray[int] that transforms the input grid to the corresponding
output grid.
Hint: You may want to use the following guidance to implement the function:
```

```
The bottom-most row that contains colored squares remains the same from test
 input to test output. Meanwhile, the other rows all start one column more
to the right in the test output compared to the test input.
The number in the input grid can be mapped to the following colors:0:black;
1:blue; 2:red; 3:green; 4:yellow; 5:grey; 6:fuschia; 7:orange; 8:teal; 9:
brown
Just reply with the implementation of transform_grid(input_grid: np.ndarray[
int]) in Python and nothing else, each cell in the output should only be
numbers from 0 to 9. Please contains the necessary import statements.
```

**Refining image-manipulating program.** We refine the codes that fail to generate correct answers by providing LLMs with previously failed codes and their output results.

This is the refinement prompt for codes that raise runtime error.

```
-----  Role: user  --------------------

Example 0:
Input:
[[0 0 0 0 0 0 0 0]
 [0 6 6 6 0 0 0 0]
 [0 6 0 0 6 0 0 0]
 [0 0 6 0 0 6 0 0]
 [0 0 0 6 0 0 6 0]
 [0 0 0 0 6 6 6 0]
 [0 0 0 0 0 0 0 0]
 [0 0 2 2 2 0 0 0]
 [0 0 2 0 0 2 0 0]
 [0 0 0 2 2 2 0 0]
 [0 0 0 0 0 0 0 0]
 [0 0 0 0 0 0 0 0]
 [0 0 0 0 0 0 0 0]
 [0 0 0 0 0 0 0 0]]
Output:
[[0 0 0 0 0 0 0 0]
 [0 0 6 6 6 0 0 0]
 [0 0 6 0 0 6 0 0]
 [0 0 0 6 0 0 6 0]
 [0 0 0 0 6 0 6 0]
 [0 0 0 0 6 6 6 0]
 [0 0 0 0 0 0 0 0]
 [0 0 0 2 2 2 0 0]
 [0 0 0 2 0 2 0 0]
 [0 0 0 2 2 2 0 0]
 [0 0 0 0 0 0 0 0]
 [0 0 0 0 0 0 0 0]
 [0 0 0 0 0 0 0 0]
 [0 0 0 0 0 0 0 0]]
Example 1:
Input:
[[0 0 0 0 0 0 0 0]
 [0 8 8 8 8 8 0 0]
 [0 8 0 0 0 8 0 0]
 [0 0 8 0 0 0 8 0]
 [0 0 0 8 0 0 0 8]
 [0 0 0 0 8 8 8 8]
 [0 0 0 0 0 0 0 0]
 [0 0 0 0 0 0 0 0]]
Output:
[[0 0 0 0 0 0 0 0]
```

```
[0 0 8 8 8 8 8 0 0]
[0 0 8 0 0 0 0 8 0]
[0 0 0 8 0 0 0 0 8]
[0 0 0 0 8 0 0 0 8]
[0 0 0 0 8 8 8 8 8]
[0 0 0 0 0 0 0 0 0]
[0 0 0 0 0 0 0 0 0]]
```

Now, please write a python program transform_grid(input_grid: np.ndarray[int
]) -> np.ndarray[int] that transforms the input grid to the corresponding
output grid.
Hint: You may want to use the following guidance to implement the function:
The bottom-most row that contains colored squares remains the same from test
 input to test output. Meanwhile, the other rows all start one column more
to the right in the test output compared to the test input.
The number in the input grid can be mapped to the following colors:0:black;
1:blue; 2:red; 3:green; 4:yellow; 5:grey; 6:fuschia; 7:orange; 8:teal; 9:
brown
Just reply with the implementation of transform_grid(input_grid: np.ndarray[
int]) in Python and nothing else, each cell in the output should only be
numbers from 0 to 9.
This is the code you wrote last time:
```
import numpy as np

def transform_grid(input_grid: np.ndarray[int]) -> np.ndarray[int]:
    nrows, ncols = input_grid.shape
    output_grid = np.zeros_like(input_grid)

    # Initialize variables to trace contiguous block of colored rows
    in_colored_block = False
    top_color_index = 0
    bottom_color_index = 0

    # Identify the contiguous main block of colored rows from bottom to top
    for idx in range(nrows - 1, -1, -1):
        if np.any(input_grid[idx] != 0):
            if not in_colored_block:
                bottom_color_index = idx
                in_colored_block = True
            top_color_index = idx
        else:
            if in_colored_block:
                break

    # Apply the transformation to the identified colored block
    shift = 0
    for r in range(bottom_color_index, top_color_index - 1, -1):
        if np.any(input_grid[r] != 0):
            output_grid[r, shift:(ncols - (ncols - ncols + shift))] =
input_grid[r, 0:(ncols - shift)]
            shift += 1

    return output_grid

```
It generates an error:
wrong output format: Error: could not broadcast input array from shape (8,)
into shape (7,)
 The correct format should be np.array
Please correct the error and generate the code.Just reply with the
implementation of transform_grid(input_grid: np.ndarray[int]) in Python and

the and nothing else, each cell in the output should only be numbers from 0
to 9. Please contains the necessary import statements.

This is the refinement prompt for codes that produce wrong answers.

```
-----  Role: user  --------------------

Example 0:
Input:
[[0 0 0 0 0 0 0 0 0]
 [0 6 6 6 0 0 0 0 0]
 [0 6 0 0 6 0 0 0 0]
 [0 0 6 0 0 6 0 0 0]
 [0 0 0 6 0 0 6 0 0]
 [0 0 0 0 6 6 6 0 0]
 [0 0 0 0 0 0 0 0 0]
 [0 0 2 2 2 0 0 0 0]
 [0 0 2 0 0 2 0 0 0]
 [0 0 0 2 2 2 0 0 0]
 [0 0 0 0 0 0 0 0 0]
 [0 0 0 0 0 0 0 0 0]
 [0 0 0 0 0 0 0 0 0]
 [0 0 0 0 0 0 0 0 0]]
Output:
[[0 0 0 0 0 0 0 0 0]
 [0 0 6 6 6 0 0 0 0]
 [0 0 6 0 0 6 0 0 0]
 [0 0 0 6 0 0 6 0 0]
 [0 0 0 0 6 0 6 0 0]
 [0 0 0 0 6 6 6 0 0]
 [0 0 0 0 0 0 0 0 0]
 [0 0 0 2 2 2 0 0 0]
 [0 0 0 2 0 2 0 0 0]
 [0 0 0 2 2 2 0 0 0]
 [0 0 0 0 0 0 0 0 0]
 [0 0 0 0 0 0 0 0 0]
 [0 0 0 0 0 0 0 0 0]
 [0 0 0 0 0 0 0 0 0]]
Example 1:
Input:
[[0 0 0 0 0 0 0 0 0]
 [0 8 8 8 8 8 0 0 0]
 [0 8 0 0 0 0 8 0 0]
 [0 0 8 0 0 0 0 8 0]
 [0 0 0 8 0 0 0 0 8]
 [0 0 0 0 8 8 8 8 8]
 [0 0 0 0 0 0 0 0 0]
 [0 0 0 0 0 0 0 0 0]]
Output:
[[0 0 0 0 0 0 0 0 0]
 [0 0 8 8 8 8 8 0 0]
 [0 0 8 0 0 0 8 0 0]
 [0 0 0 8 0 0 0 8 0]
 [0 0 0 0 8 0 0 0 8]
 [0 0 0 0 8 8 8 8 8]
 [0 0 0 0 0 0 0 0 0]
 [0 0 0 0 0 0 0 0 0]]

Now, please write a python program transform_grid(input_grid: np.ndarray[int
]) -> np.ndarray[int] that transforms the input grid to the corresponding
output grid.
```

Hint: You may want to use the following guidance to implement the function:
The bottom-most row that contains colored squares remains the same from test
 input to test output. Meanwhile, the other rows all start one column more
to the right in the test output compared to the test input.
The number in the input grid can be mapped to the following colors:0:black;
1:blue; 2:red; 3:green; 4:yellow; 5:grey; 6:fuschia; 7:orange; 8:teal; 9:
brown
Just reply with the implementation of transform_grid(input_grid: np.ndarray[
int]) in Python and nothing else, each cell in the output should only be
numbers from 0 to 9.

```
import numpy as np

def transform_grid(input_grid: np.ndarray[int]) -> np.ndarray[int]:
    nrows, ncols = input_grid.shape
    output_grid = np.zeros_like(input_grid)

    # Finding the bottom-most row with non-zero entries
    for row in range(nrows-1, -1, -1):
        if np.any(input_grid[row] != 0):
            last_colored_row = row
            break

    # Copy the last colored row as is:
    output_grid[last_colored_row] = input_grid[last_colored_row]

    # Shift each row above the bottom-most colored row one column to the
right
    for row in range(last_colored_row-1, -1, -1):
        if np.any(input_grid[row] != 0):
            output_grid[row, 1:] = input_grid[row, :-1]

    return output_grid

```

Task 1:
Input:
 [[0, 0, 0, 0, 0, 0, 0, 0, 0], [0, 6, 6, 6, 0, 0, 0, 0, 0], [0, 6, 0, 0, 6,
0, 0, 0, 0], [0, 0, 6, 0, 0, 6, 0, 0, 0], [0, 0, 0, 6, 0, 0, 6, 0, 0], [0,
0, 0, 0, 6, 6, 6, 0, 0], [0, 0, 0, 0, 0, 0, 0, 0, 0], [0, 0, 2, 2, 2, 0, 0,
0, 0], [0, 0, 2, 0, 0, 2, 0, 0, 0], [0, 0, 0, 2, 2, 2, 0, 0, 0], [0, 0, 0,
0, 0, 0, 0, 0, 0], [0, 0, 0, 0, 0, 0, 0, 0, 0], [0, 0, 0, 0, 0, 0, 0, 0, 0],
 [0, 0, 0, 0, 0, 0, 0, 0, 0]]
Correct Output:
 [[0, 0, 0, 0, 0, 0, 0, 0, 0], [0, 0, 6, 6, 6, 0, 0, 0, 0], [0, 0, 6, 0, 0,
6, 0, 0, 0], [0, 0, 0, 6, 0, 0, 6, 0, 0], [0, 0, 0, 0, 6, 0, 6, 0, 0], [0,
0, 0, 0, 6, 6, 6, 0, 0], [0, 0, 0, 0, 0, 0, 0, 0, 0], [0, 0, 0, 2, 2, 2, 0,
0, 0], [0, 0, 0, 2, 0, 2, 0, 0, 0], [0, 0, 0, 2, 2, 2, 0, 0, 0], [0, 0, 0,
0, 0, 0, 0, 0, 0], [0, 0, 0, 0, 0, 0, 0, 0, 0], [0, 0, 0, 0, 0, 0, 0, 0, 0],
 [0, 0, 0, 0, 0, 0, 0, 0, 0]]
Code Output:
 [[0, 0, 0, 0, 0, 0, 0, 0, 0], [0, 0, 6, 6, 6, 0, 0, 0, 0], [0, 0, 6, 0, 0,
6, 0, 0, 0], [0, 0, 0, 6, 0, 0, 6, 0, 0], [0, 0, 0, 0, 6, 0, 0, 6, 0], [0,
0, 0, 0, 6, 6, 6, 0, 0], [0, 0, 0, 0, 0, 0, 0, 0, 0], [0, 0, 0, 2, 2, 2, 0,
0, 0], [0, 0, 0, 2, 0, 0, 2, 0, 0], [0, 0, 0, 2, 2, 2, 0, 0, 0], [0, 0, 0,
0, 0, 0, 0, 0, 0], [0, 0, 0, 0, 0, 0, 0, 0, 0], [0, 0, 0, 0, 0, 0, 0, 0, 0],
 [0, 0, 0, 0, 0, 0, 0, 0, 0]]
Task 2:
Input:
 [[0, 0, 0, 0, 0, 0, 0, 0, 0], [0, 8, 8, 8, 8, 8, 0, 0, 0], [0, 8, 0, 0, 0,
0, 8, 0, 0], [0, 0, 8, 0, 0, 0, 0, 8, 0], [0, 0, 8, 0, 0, 0, 0, 0, 8], [0,

```
0, 0, 0, 8, 8, 8, 8, 8], [0, 0, 0, 0, 0, 0, 0, 0, 0], [0, 0, 0, 0, 0, 0, 0,
0, 0]]
Correct Output:
 [[0, 0, 0, 0, 0, 0, 0, 0, 0], [0, 0, 8, 8, 8, 8, 8, 0, 0], [0, 0, 8, 0, 0,
0, 0, 8, 0], [0, 0, 0, 8, 0, 0, 0, 0, 8], [0, 0, 0, 0, 8, 0, 0, 0, 8], [0,
0, 0, 8, 8, 8, 8, 8], [0, 0, 0, 0, 0, 0, 0, 0, 0], [0, 0, 0, 0, 0, 0, 0,
0, 0]]
Code Output:
 [[0, 0, 0, 0, 0, 0, 0, 0, 0], [0, 0, 8, 8, 8, 8, 8, 0, 0], [0, 0, 8, 0, 0,
0, 0, 8, 0], [0, 0, 0, 8, 0, 0, 0, 0, 8], [0, 0, 0, 0, 8, 0, 0, 0, 0], [0,
0, 0, 8, 8, 8, 8, 8], [0, 0, 0, 0, 0, 0, 0, 0, 0], [0, 0, 0, 0, 0, 0, 0,
0, 0]]

Please correct the error and generate the code. Just reply with the
implementation of transform_grid(input_grid: np.ndarray[int]) in Python and
nothing else, each cell in the output should only be numbers from 0 to 9.
Please contains the necessary import statements.
```

## A.7 Example trees generated by *REx*

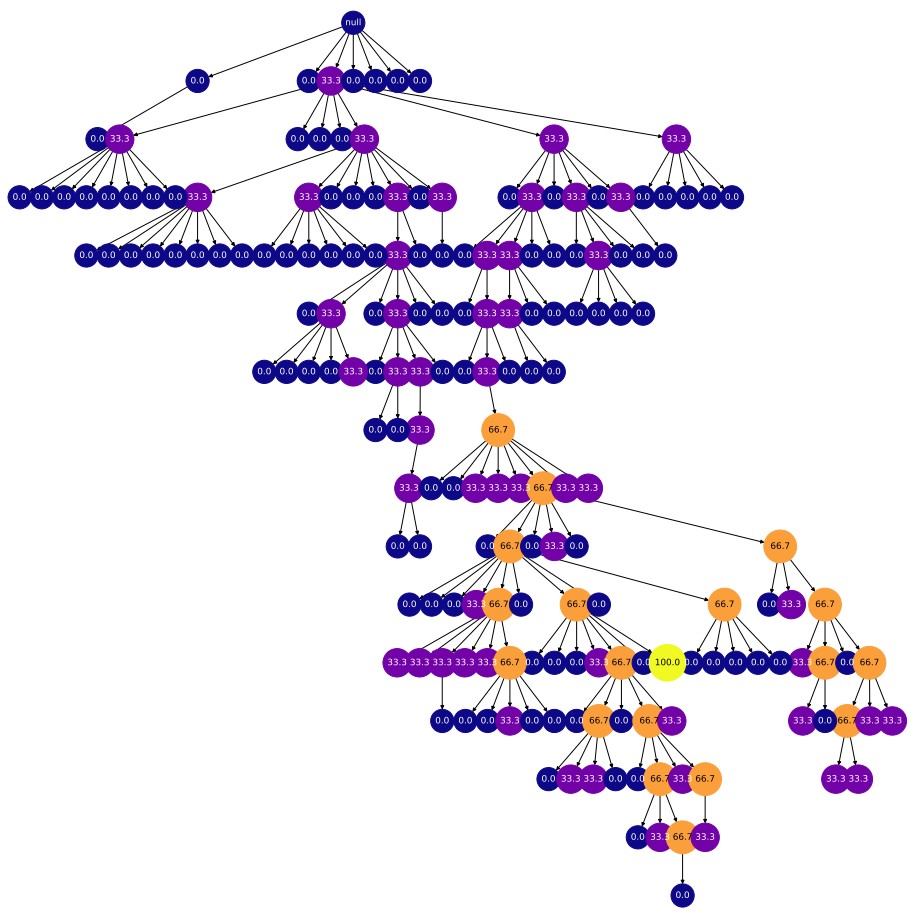

Figure 14: Search Tree

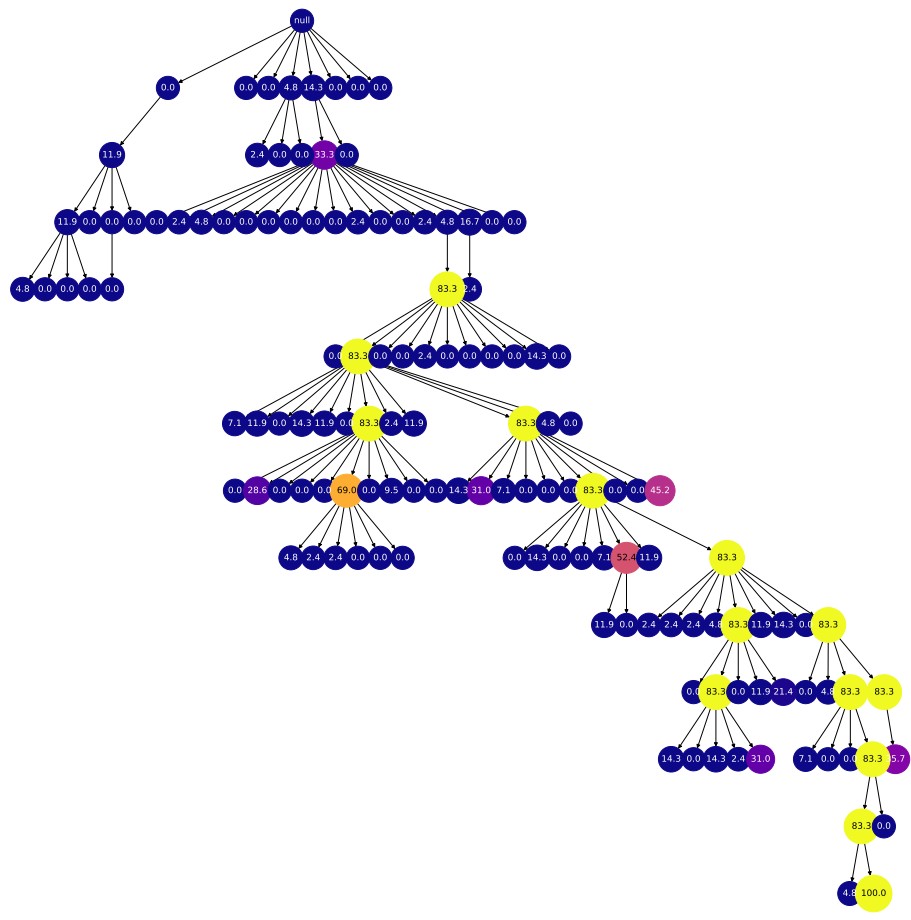

Figure 15: Search Tree

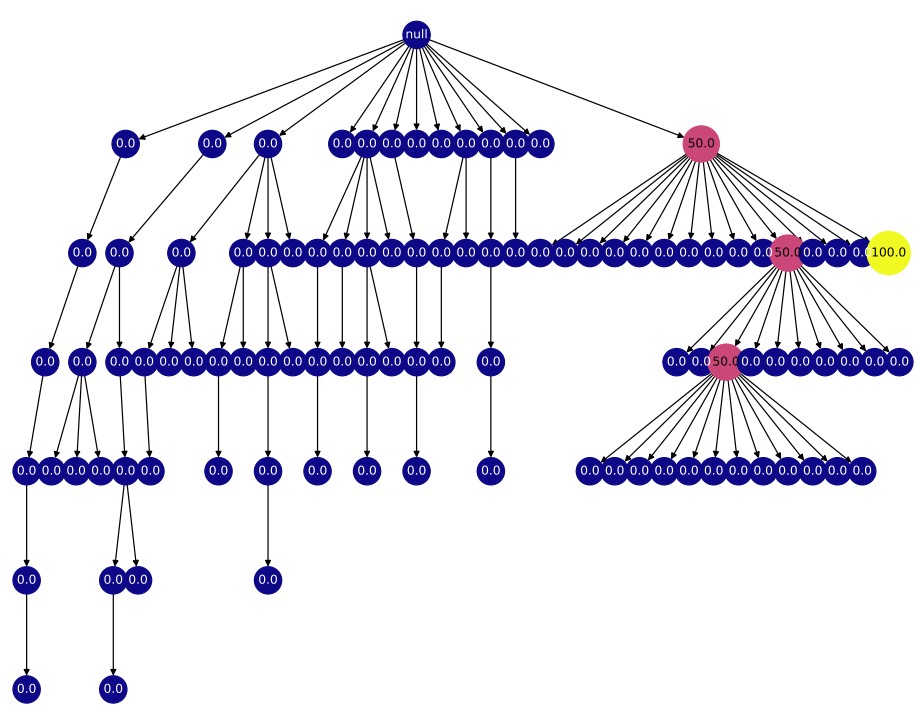

Figure 16: Search Tree

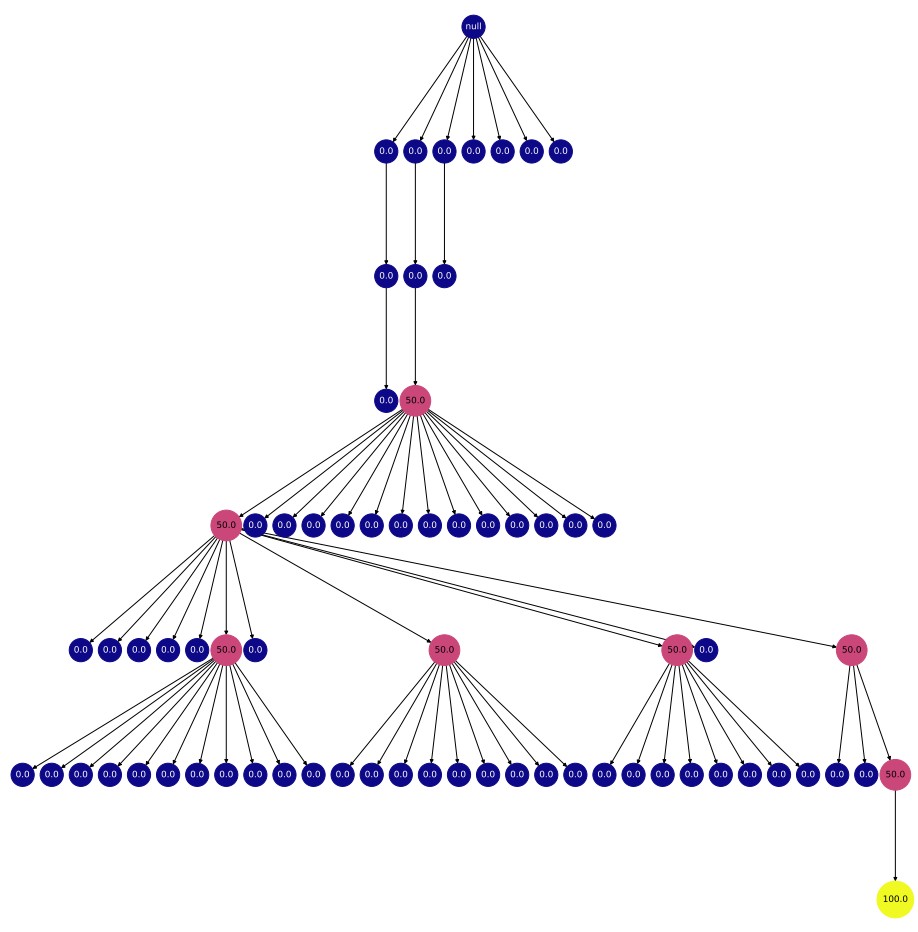

Figure 17: Search Tree

