# OpenReview forum: "Code Repair with LLMs gives an Exploration-Exploitation Tradeoff"
_NeurIPS.cc/2024/Conference — NeurIPS 2024 poster_

### Official Review · Reviewer_UuYA · 2024-07-11

**Soundness:** 3
**Presentation:** 3
**Contribution:** 2
**Rating:** 5
**Confidence:** 4

**Summary:**

This paper studies the code refinement problem: given a candidate program and the reason why the program fails to satisfy the user specification, an LLM is called to generate an improved program. The paper frames this as an arm-acquiring bandit problem and solves this using Thompson sampling. The evaluation is performed on three code generation tasks: competitive programming, visual program reasoning, and loop invariant generation. The results show that while the proposed approach solves only moderately more tasks than baseline given a large-compute limit, it is significantly faster to achieve a certain performance.

**Strengths:**

The paper is very well-written and easy-to-follow. I thank the authors for providing the details regarding the evaluation tasks and the prompts. Moreover, I appreciate the paper’s message on emphasizing the importance of developing better base models over constructing better refinement strategies. While it sounds like a negative result for the paper, it is definitely useful for the community.

**Weaknesses:**

REx seems like a straightforward application of Thompson sampling. Moreover, it only gives moderately improvement over baseline search methods in a large-compute setting. I do appreciate simplicity when it is combined with effectiveness, but this is not the case for REx. Therefore, I consider the technical contribution of this paper rather thin.

Moreover, a major selling point of the paper is to study an exploration-exploitation tradeoff. While the paper provides some preliminary study at Lines 179-182, I believe a more thorough analysis is needed for full understanding.

Another limitation, as also pointed out by the paper, is that only GPT-4 is studied. Would REx give better results in the large-compute setting when the self-repairing capability of the base model is weaker and the search strategy might play a more important role?

**Questions:**

Please consider addressing the points raised in the “Weakness” section.

**Limitations:**

The paper has sufficiently addressed the points concerning limitations and societal impact.

---

> ### Author Rebuttal · Authors · 2024-08-06
>
> Thank you for the review and for the suggestions. We have run new experiments that we really believe can address your concerns. Please see below.
>
> > only GPT-4 is studied. Would REx give better results in the large-compute setting when the self-repairing capability of the base model is weaker and the search strategy might play a more important role?
>
> Thank you for this idea. We ran last week other (weaker) language models, and discovered that the advantage of REx is sometimes much more pronounced: For example, on GPT3.5/APPS-competition, our method solves about ~40% more problems (relative improvement) over the second best approach.
>
> Thank you again for suggesting this, which we think significantly strengthens the paper by showing more models as well as incidentally discovering cases where REx makes a much bigger difference than we showed in the original submission.
>
> > Moreover, it only gives moderately improvement over baseline search methods in a large-compute setting. I do appreciate simplicity when it is combined with effectiveness, but this is not the case for REx. Therefore, I consider the technical contribution of this paper rather thin
>
> We hope that you can reconsider this in light of the above result from the experiment that you suggested. We also hope one can see technical contribution in cost reduction and increased hyperparameter robustness.
>
> > Moreover, a major selling point of the paper is to study an exploration-exploitation tradeoff. While the paper provides some preliminary study at Lines 179-182, I believe a more thorough analysis is needed for full understanding.
>
> We'll include videos we made recently of REx searching through refinement trees, which clearly show a balancing of exploring and exploiting. We also gave a theoretical (mathematical) analysis in L120-L132/Fig 2.
>
> Please let us know if you have further questions.

---

> > ### Comment · Reviewer_UuYA · 2024-08-08
> > **Thanks for the rebuttal**
> >
> > Thanks for running the experiments! The rebuttal has addressed my main concerns. Therefore, I am raising my score from 4 to 5. I hope the authors could consolidate the experiments on more LLMs, the application of REx to more challenging tasks, and the analysis of the exploration-exploitation tradeoff. Including these results would help strengthen the paper.

---

### Official Review · Reviewer_ikA5 · 2024-07-11

**Soundness:** 3
**Presentation:** 2
**Contribution:** 2
**Rating:** 5
**Confidence:** 3

**Summary:**

This paper explores a variety of code generation tasks, taking the approach of using a LLM to generate solutions, and conditioning the generation of each solution on the repair of a previously generated solution. It frames this as an arm-acquiring bandit problem, where each solution generated is an arm, and pulling an arm means refining a previous solution with an LLM (specifically, with GPT-4). The paper applies Thompson Sampling to this problem; specifically the paper's method is to use a heuristic-driven prior for which solution to refine at each step (i.e. the fraction of test cases passed by that solution), and then to penalize a solution each time its refinement does not lead to a perfect solution. The method is conceptually simple, and results in stronger performance (i.e. more problems solved; fewer model calls used) than a few baseline approaches.

**Strengths:**

The method presented is conceptually simple, and is easy to implement and adapt to new tasks. Therefore this method can readily be put into practice by practitioners. (However, see weakness no. 1).

The connection to bandit problems is sound, and potentially a valuable framing of the problem. (However, see weakness no. 3.) With regard to significance, making this connection could suggest a wide range of approaches from bandit literature. That said the paper does not explore this potential benefit of the framing beyond its application of Thompson Sampling.

The range of tasks considered is another strength of the paper, and a key strength of the paper is that the method demonstrates performance improvements on three of the four tasks (the exception being the easiest APPS problems). The variance of the results on NLI and ARC is high though, so while the method is an improvement, the improvement is not substantial or guaranteed.

The modesty/honesty of the limitations section is refreshing too, providing clear commentary on the biggest limitations of the work.

**Weaknesses:**

I noted as the first strength listed that the method can readily be put into practice by practitioners, but I must couch this assessment: the problem statement and method assume access to an efficient checker for whether the task is solved, which restricts the space of tasks to which the method can be applied.

A weakness of the paper is that only GPT-4 is tested. As the paper acknowledges, GPT-4 is among the strongest models for editing code. It is also among the more expensive models to run. By only evaluating the method with GPT-4, we are left to wonder about the cost tradeoffs of using smaller cheaper models with more samples vs this large model with fewer samples. Since one of the stated goals of the paper is to reduce cost (the paper states this as the goal of reducing the number of model calls), this would be valuable to understand. Since the results are high variance and of modest effect size, understanding how the method's performance varies across models would be valuable.

It is not clear that single-sample-refinement is the best framing of the problem; there are lots of ways to prompt a model beyond refinement from a single prior solution, and the paper does nothing to suggest that single-solution-refinement is a promising strategy. I think this is the biggest weakness of the problem statement; it assumes a rather rigid set of ways to apply LLMs to solving these programming tasks, and then optimizes within that rigid constraint, without justifying adequately that it is a valuable constraint.

---

Overall, given the constraints of the problem statement, the method presented is a step forward in terms of performance, formulating an idea from bandits for this program synthesis setting and achieving improved results. A key aspect of the paper's contribution is that the paper frames the search tree as infinite width and infinite depth; I cannot meaningfully comment of the novelty of this framing, though given the limited search budgets of a few hundred LLM calls the claim of infinite width and infinite depth doesn't seem critical. The paper is written clearly and the method presented is simple to follow and implement. The main restriction on the significance of the work is that the constraints made by the problem statement -- that the method works by iteratively refining a single preexisting solution at a time -- are overly constraining and not representative of the range of ways in which people apply LLMs today.

**Questions:**

In the related work section you state that it is not possible to apply MCTS or any of its standard variants to the problem statement out-of-the-box. Could you elaborate on (or refine) this position, and comment on whether small modifications would be sufficient to apply MCTS to this problem?

Since the LLM is shown the full specification (Page 3, eq (4)), there is risk of it generating solutions that satisfy the specification literally without generalizing to satisfy the intent of the specification. A simple example of this would be if the model writes special case handling for each of the constraints in the specification, rather than capturing the underlying problem to be solved. Do you observe this in practice at all?

How do methods that fit the problem statement, i.e. methods that operate by only refining one solution at a time to produce new solutions, compare with other approaches to using LLMs to solve these tasks. Showcasing what other approaches have been applied and how they perform, or alternatively making a strong case that this particular formulation is a good choice, would improve the paper.

At Line 92 you consider an alternative heuristic of edit-distance to a target output. Am I correct in understanding that having a target output available for the heuristic would obviate the need for applying the method?

One limitation of the approach is that as information comes in about proposed solutions, the system does not use that to learn anything about other previous solutions, even if they are heavily related. Similar to weakness 3 and question 3, approaches that leverage this source of information sit outside the problem statement. Could you comment on the relevance of such approaches, either comparing with them or justifying their absence?

nit: Figure 1, Right: I would encourage you to label the outgoing edges "exploit" and "explore" rather than labeling the nodes or incoming edges as is currently done. This is because the current figure makes it look as if generating labelled nodes was done as exploitation vs exploration, where in fact it is refining those nodes that would be exploitation or exploration.

**Limitations:**

The authors address the limitations of the work in Section 7, Limitations. They explain the effect of only testing their approach on GPT-4, and discuss the limited effect size of their approach at solving more problems overall.

---

> ### Author Rebuttal · Authors · 2024-08-06
>
> Thank you for the helpful review! Please see below for our responses, and see the global response PDF for new experimental results motivated by your suggestions.
>
> > only GPT-4 is tested
>
> Thanks or the suggestion of testing other models. Please see the global response PDF for new results on other LLMs. Although the results are still coming in, and the absolute numbers are somewhat different, the overall qualitative conclusion is the same, with one exception: We've discovered that our method seems to help a lot with cheaper models, which means that it might be more broadly applicable then we originally sold it as.
>
> > you state that it is not possible to apply MCTS... Could... small modifications would be sufficient to apply MCTS to this problem?
>
> While such modifications could, in principle, likely be invented, they would probably be more complicated than REx, which is considerably simpler than MCTS, despite capturing MCTS-esque dynamics. However, we'd be happy to try any specific modifications you suggest next week during the discussion period.
>
> > Since the LLM is shown the full specification... there is risk of it generating solutions that satisfy the specification literally without generalizing to satisfy the intent of the specification
>
> We evaluate on holdout inputs for ARC. For loop invariants we use a formal verifier to check that the invariant actually holds for all possible inputs. APPS, unfortunately, does not come with designated holdout tests, and conventionally has been evaluated without them [e.g. Olausson ICLR '24; Zelikman NeurIPS '23]. However, APPS averages 27.6 tests/problem, so it would be very hard to "overfit" without recovering the true algorithm, and in manually inspecting 15 program solutions we observe no memorization of isolated test-cases. The revision will mention all these issues.
>
> > other [non-refinement] approaches to using LLMs to solve these tasks
>
> We are happy to try any particular baseline that you suggest next week during the discussion period. We do think that refinement is an especially popular and simple approach, though, so we think focusing on refinement is a reasonable research strategy for this paper.
>
> > Am I correct in understanding that having a target output available for the heuristic would obviate the need for applying the method?
>
> We'd still need to apply our method because we would still need to generate a program that maximizes the heuristic value.
>
> > One limitation of the approach is that as information comes in about proposed solutions, the system does not use that to learn anything about other previous solutions, even if they are heavily related. Similar to weakness 3 and question 3, approaches that leverage this source of information sit outside the problem statement. Could you comment on the relevance of such approaches, either comparing with them or justifying their absence?
>
> This would be a fascinating direction to explore, for example by incorporating a distance metric between programs and using probabilistic kernel methods for the prior (such as a Gaussian process). But this requires a metric between program source code in order to determine what programs are similar, which introduces its own complexities and would likely be more brittle and sensitive to hyperparameters and/or require in-domain training data. Still, it could be an interesting direction to explore, and we will add it to the future work.
>
> Thanks again for the review and for the suggestions of future work. Please let us know if you have any further questions.

---

> > ### Comment · Reviewer_ikA5 · 2024-08-13
> >
> > Thanks for your rebuttal. The inclusion of additional language models makes the results more compelling.
> >
> > A key remaining concern is that it is not clear that single-sample-refinement is the best framing of the problem; there are lots of ways to prompt a model beyond refinement from a single prior solution, and the paper does nothing to suggest that single-solution-refinement is a promising strategy compared to these alternatives.
> >
> > Here are a couple examples of categories of alternatives: single-sample prompting techniques like scratchpad/Chain-of-Thought and all its variants, LLM-chain approaches that generate a single result, like ReAct and its variants (one can view these as improvements on how to refine, rather than what to refine), and refinement approaches like EvoPrompting that consider more than one sample at a time.

---

### Official Review · Reviewer_rHDY · 2024-07-13

**Soundness:** 3
**Presentation:** 3
**Contribution:** 3
**Rating:** 6
**Confidence:** 4

**Summary:**

This paper proposes to improve the iterative code refinement process by prioritizing the “good” programs, where the goodness is defined by a heuristic estimator -- the program that passes the more test cases is better. To balance exploration (explore a lesser refined program) and exploitation (refine the best program), the paper formulates this problem as a bandit problem, and solves with Thompson Sampling. The proposed method achieves modest but consistent improvements across various program synthesis benchmarks (loop invariant, visual transformation program, competition problems).

**Strengths:**

- The paper’s formulation of the iterative program refinement problem is insightful.
- The exploration-exploitation tradeoff is an important yet surprisingly overlooked aspect for effective code repair. This paper systematically analyzes different search policies.
- The proposed solution with bandit algorithms is very simple yet effective, and seems to be robust across various repair benchmarks.

**Weaknesses:**

### **Technique**

The scope of this paper is limited to solving small, isolated programming challenges given a relatively large compute limit. It is unclear whether such techniques can be applied to more realistic settings (where we cannot afford large numbers of full program samples). For example, to repair a bug in a repository, it might be more important to improve the code refinement step itself, either with better prompting or training.

It is also unclear whether similar results can be achieved using more advanced prompting techniques (e.g., few-shot or CoT).

Formulating the problem as a bandit problem seems a little unnatural, as the algorithm terminates as soon as a positive reward is observed. In other words, the accumulated reward is always 0 before termination. It’d be great if the paper can include more theoretical discussion regarding how this special assumption would affect the theoretical guarantee of the algorithm.

### **Evaluation**
The improvement of REx seems marginal.
- Compared with alternative search algorithms, REx often brings only marginal improvements given a large enough budget (Figure 4).
- Also on the some dataset (loop invariant), BFS (with the optimal hyperparameter) is even better than REx with smaller sampling budget.

The authors should list the cost of baselines for fair comparison.
- In Figure 4, the existing LLM-based methods should be marked as data points in the plot, with their corresponding sample budget (if they use the same base LM).


According to Table 1 in appendix, the Greedy baseline seems to be very strong, even though the paper only considered two values for its single hyperparameter. This leads me to wonder whether such a simple baseline has more potential.
- The hyperparameter, specifically the heuristic value of the initial program, controls when to resample a new program instead of refining existing ones. This seems to be highly problem-specific. Instead of having a fixed value for all problems, can we maintain a (moving) average of all sampled solutions for one particular problem?

Also, each arm is not independent, as the refined program $\rho^\prime$ is correlated with the original program $\rho$.
- I am curious about whether it is possible to better incorporate the observations of the refined program(s) to update the posterior belief of the original program. Although the current reward (refined program $\rho^\prime$ satisfies all input-output pairs) is very sparse, we may use the heuristic metric $h(\rho^\prime)$ as an alternative.
- In fact, MAB has been applied for fuzz testing [a,b], where code mutation is performed instead of code refinement. [a] formulates the problem as hierarchical multi-armed bandits, and [b] also formulates the problem as arm-acquiring bandits. It could be an interesting line of related work to check.

[a] Ran, Dezhi, et al. "Badge: prioritizing UI events with hierarchical multi-armed bandits for automated UI testing." 2023 IEEE/ACM 45th International Conference on Software Engineering (ICSE). IEEE, 2023.

[b] Yang, Chenyuan, et al. "White-box compiler fuzzing empowered by large language models." arXiv 2023



### **Implementation**

Some implementation details are unclear. For example, what is the temperature for ARC and loop invariant experiments? Appendix A5 lists temperature=1.0 for APPS, is it the same for all considered benchmarks?

### **Minor**

Formula (11):  $2+C+N_\rho$ should be $2+2C+N_\rho$

Figure-6: The x-axis is somewhat unclear. Does it represent the number of requests needed to achieve performance comparable to the best baselines? Please clarify.

**Questions:**

Could you please explain and address the weaknesses?

**Limitations:**

The limitations are well addressed.

---

> ### Author Rebuttal · Authors · 2024-08-06
>
> Thank you for the thoughtful input and for your support. Please see below our responses.
>
> > REx often brings only marginal improvements given a large enough budget (Figure 4)
>
> We agree! REx isn't magic: It is simply a more hyperparameter-robust, cost-saving refinement policy that also modestly improves the number of solved problems overall. We're excited about REx because it seems to help generically across the board, so we're optimistic that it could have an impact for many researchers, and why we've emphasized just how easy it is to implement in Python. (But please see below our response for the next comment on where we've taken REx recently, and the global response for some cases where REx makes a big difference when using cheap models)
>
> > The scope of this paper is limited to solving small, isolated programming challenges given a relatively large compute limit
>
> We have since applied REx to much harder problems involving programs with hundreds of lines of code. We will reference these results in the camera ready. In fact, the original reason we considered a relatively large compute limit was because we believe that these harder problems require many more rounds of refinement.
>
> > the Greedy baseline seems to be very strong, even though the paper only considered two values for its single hyperparameter
>
> We originally did this because greedy's hyperparameter is unique in that it has very few reasonable settings, as it corresponds to the heuristic value of an empty program. Setting it to e.g. 1 would mean that the search policy would never even try doing refinement; setting it close to zero means that it would only ever refine the initial program once. To double-check these intuitions we will rerun the greedy experiments with more hyperparameters.
>
> > Also on the some dataset (loop invariant), BFS (with the optimal hyperparameter) is even better than REx with smaller sampling budget.
>
> The important point here is that you can't count on BFS always being the best: on a different dataset (APPS), it's actually the worst performing method! We want methods that are robust across hyperparameters and robust across datasets, which means acknowledging that sometimes special hyperparameters can make a method *seem* superior on a particular datasets for a particular sampling budgets. For loop invariants in particular, the box and whisker plots show performance across a range of hyperparameters, for which BFS actually tends to be worse than REx (Fig 4/6).
>
> > It is also unclear whether similar results can be achieved using more advanced prompting techniques (e.g., few-shot or CoT).
>
> REx is orthogonal to the prompting technique: Given a refinement prompt (e.g., few-shot or CoT), REx then works through an outer loop that repeatedly uses that prompt.
>
> > what is the temperature for ARC and loop invariant experiments? Appendix A5 lists temperature=1.0 for APPS, is it the same for all considered benchmarks?
>
> Yes, temperature=1 for all considered benchmarks.
>
> > In fact, MAB has been applied for fuzz testing [a,b]...
>
> That work is indeed very related! We will cite and discuss the papers you mention.
>
> Thank you for your review and for your support. Please let us know if we can answer further questions.

---

> > ### Comment · Reviewer_rHDY · 2024-08-08
> > **Thanks for the rebuttal**
> >
> > Thanks for answering all my questions! I'll be keeping my score of 6 and support the acceptance of the paper.

---

### Official Review · Reviewer_mETF · 2024-07-30

**Soundness:** 3
**Presentation:** 4
**Contribution:** 3
**Rating:** 7
**Confidence:** 4

**Summary:**

The paper identifies that LLM refinement process can be formulated as (arm-acquiring) non-contextual bandit problem which can be solved optimally (in the limit) using principled bandit-algorithms like Thompson Sampling against heuristic based solutions. It applies this idea to three code refinement tasks and demonstrates empirical improvements from their approach.

**Strengths:**

1. **Novelty**. The proposed formulation although simple is novel in the context of multi-turn LLM applications and provides insights discovered beyond existing works. The proposed solution using Thompson-Sampling with Beta priors is straightforward but effective at improving efficiency.

2. **Writing and Clarity**. The paper was pleasant to read and provides appropriate context connecting prior works. There is some missing information detailed in weaknesses.

3. **Results**. Comprehensive experiments across three (widely varying tasks) demonstrating empirical effectiveness over simpler heuristic based exploration-exploitation baselines.

**Weaknesses:**

1. **Non-contextual Bandit formulation**. While the bandit formulation is clean and simple, it leaves more to be desired and makes simplifying assumptions. Particularly, since the bandit formulation neither contextualizes on the problem statement nor considers "depth" of the arm as a special attribute. This seems different from expected behavior

    1.1 **Problem Statement**. Problem statement links is directly linked with the complexity of task. For very challenging tasks, optimal strategies would differ from easy tasks, where for challenging tasks more exploration across "width" might be expected.
    1.2 **Depth of Arm**. It might be reasonable to have different priors for "arms" at different "depths". For example, pulling the "empty program arm" seems special compared against other arms.

At the same time, without explicitly modeling these contextual components the proposed approach seems strong -- perhaps accounted for by more "diverse" hyper-parameter search space (discussed further in Questions section point 1.).

2. **Choice of LLM**. As authors mentioned, choice of the LLM might play a role in the performance of this work. It might be useful to study this axis further and see if the findings change by taking a weaker model like `gpt-3.5-turbo`.

3. **Invariant Task Grading**. The invariant task is a core task studied in the paper and the authors point out that they check
> being a sufficiently strong inductive invariant (i.e., precondition, induction, and postcondition).

The statement is not precise enough and needs to be further clarified. Additionally, is it possible for models to generate trivial invariants (e.g. - `True is True`) and fool the grader?

4. **Contamination**. APPS benchmark is considerably older and released before the cutoff date for newer GPT-4 models (authors do not specify the GPT-4 version used here).

5. **Missing details**. There are some missing/hard-to-find experimental details in the Appendix. Particularly, what feedback is used for refinement of invariant task based on the types of mistakes it makes -- weak invariant, failing invariant, solver timeout.

**Questions:**

1. Hyper-parameter variance. It seems that optimal hyper-parameters vary considerably across tasks (Table 1 in Appendix). Perhaps, it can be viewed that the hyper-parameter choice C allows better control over the exploration-exploitation search space allowing it to exhibit widely different behaviors -- accounting for the lack of contextual features discussion in weaknesses point 1?

2. **Connecting the Appendix**. There are many interesting discussions and details in the Appendix (e.g. A.2) etc which are not connected in the main paper. I would recommend the authors to add them (and at a minimum link) to the main paper. Similarly, lot of experimental settings are scattered throughout the Appendix which should be linked from the main paper via forward references making them more accessible.

3. **Figure 3**. Figure three while highlights the tasks considered is perhaps considerably larger and authors can truncate the examples and provide more experimental details and results in the main paper.

**Limitations:**

Thoughtful discussion on limitations of the approach in a.) allowing solving more problems and b.) differences arising from the choice of LLM is already provided.

---

> ### Author Rebuttal · Authors · 2024-08-06
>
> Thank you for the feedback and the supportive review. Below we answer your main questions.
>
> > Choice of LLM. As authors mentioned, choice of the LLM might play a role in the performance of this work. It might be useful to study this axis further
>
> Great idea! In the global response we have attached a PDF showing results for GPT3.5, Llama, and Claude. Although the absolute numbers are different, the qualitative outcomes are highly similar, and in fact, the advantage of REx seems larger for gpt3.5.
>
> > Non-contextual Bandit formulation. While the bandit formulation is clean and simple, it leaves more to be desired and makes simplifying assumptions. Particularly, since the bandit formulation neither contextualizes on the problem statement nor considers "depth" of the arm as a special attribute. This seems different from expected behavior
>
> This would be interesting to explore. Conditioning on depth or program/problem embeddings could be very promising, but would introduce more free parameters. We’ll add it to the discussion section as a future direction.
>
> > The invariant task is a core task studied in the paper and the authors point out that they check "being a sufficiently strong inductive invariant (i.e., precondition, induction, and postcondition)." The statement is not precise enough and needs to be further clarified. Additionally, is it possible for models to generate trivial invariants (e.g. - True is True) and fool the grader?
>
> It is not possible to "fool" the verifier with trivial invariants, which we will clarify by revising the main text to explain precondition/induction/postcondition. Briefly the *precondition* means that the invariant holds at the beginning of the loop; *induction* means when the invariant is true it stays true after another loop iteration; and *postcondition* means if the invariant is true and the loop terminates then the assertions after the loop are satisfied. `True is True` would not satisfy the postcondition constraint.
>
> > Contamination. APPS benchmark is considerably older and released before the cutoff date for newer GPT-4 model
>
> We'll revise to mention that this as an important concern for APPS. Originally we (incorrectly) assumed that, given the popularity of benchmarking GPT4 on APPS, there must be good reason to assume no contamination, but upon digging through the GPT4 tech report this week, we could find no such justification. Good catch.
>
> In the global response we show new results on GPT3.5, which deriving from GPT3 is a lot less likely to have seen APPS in pretraining.
>
> > It seems that optimal hyper-parameters vary considerably across tasks (Table 1 in Appendix). Perhaps, it can be viewed that the hyper-parameter choice C allows better control over the exploration-exploitation search space allowing it to exhibit widely different behaviors -- accounting for the lack of contextual features discussion in weaknesses point 1?
>
> Yes, it's possible that with a richer set of features for the context, there could be a universal optimal hyperparameter setting. We wanted to explore many hyper parameters to understand sensitivity of different methods, and REx was almost always the best both in terms of median and max hyperparameter settings.
>
> > Missing details. There are some missing/hard-to-find experimental details in the Appendix. Particularly, what feedback is used for refinement of invariant task based on the types of mistakes it makes... There are many interesting discussions and details in the Appendix (e.g. A.2) etc which are not connected in the main paper. I would recommend the authors to add them (and at a minimum link) to the main paper. Similarly, lot of experimental settings are scattered throughout the Appendix which should be linked from the main paper via forward references making them more accessible.
>
> Thank you for suggesting these improvements. We are revising to incorporate this feedback, especially adding links to the appendix from the main text.
>
> Thanks again for your helpful feedback and please let us know if there are any further questions we can answer.

---

> > ### Comment · Reviewer_mETF · 2024-08-09
> >
> > Thanks for answering my questions. I will maintain my rating.

---

### Author Rebuttal · Authors · 2024-08-07

Multiple reviewers raised the concern that we only evaluated our method on GPT4.
To address this concern we are in the process of running our experiments on GPT3.5, Llama3, and Claude3. The attached PDF shows preliminary in-progress results.

Although the results are still preliminary, the advantage of REx seems to like it might be more pronounced for some weaker models such as gpt3.5: Speculatively, this may be due to a "saturating" effect when the model is powerful enough. We think this makes the work stronger because it reveals a regime that we have not discovered previously where our method especially shines. Thank you to the reviewers who suggested these experiments.

---

### Decision · Program_Chairs · 2024-09-25

**Decision:**

Accept (poster)

**Comment:**

This paper proposes tackling the search process of (multi-turn) code repair as an armed bandit. In an evaluation of multiple tasks and a few LLMs this method shows to perform mildly better than baselines reducing the cost for achieving similar results to a baseline.
This method seems useful to the community, seems relatively simple to implement, and achieves tangible improvements. While additional information can be used by the banding (_e.g._ depth) and further improvements can be made, this technique is technically sound and hence all the reviewers and myself believe that there is no reason to reject this work.

I would kindly ask the authors to strongly consider incorporating some of the additional ablations suggested by reviewer `ikA5` in their camera-ready.